

# Modelling GNSS-observed seasonal velocity changes of the Ross Ice Shelf, Antarctica, using the Ice-sheet and Sea-level System Model (ISSM)

Francesca Baldacchino[1,2], Nicholas R. Golledge[1], Huw Horgan[1,3,4], Mathieu Morlighem[5], Alanna V. Alevropoulos-Borrill[1], Alena Malyarenko[6,1], Alexandra Gossart[1], Daniel P. Lowry[7], and Laurine van Haastrecht[1]

[1]Te Puna Pātiotio | Antarctic Research Centre, Te Herenga-Waka, Victoria University of Wellington, Aotearoa | New Zealand
[2]Institute of Geodesy, Graz University of Technology, Graz, Austria
[3]Laboratory of Hydraulics, Hydrology and Glaciology (VAW), ETH Zurich, Zurich, Switzerland
[4]Swiss Federal Institute for Forest, Snow and Landscape Research (WSL), Birmensdorf, Switzerland
[5]Department of Earth Sciences, Dartmouth College, Hanover, NH 03755, USA
[6]School of Earth and Environment | Te Kura Aronukurangi, University of Canterbury | Te Whare Wānanga o Waitaha, Ōtautahi | Christchurch
[7]Department of Surface Geosciences, GNS Science, Lower Hutt, New Zealand

**Correspondence:** Francesca Baldacchino (francesca.baldacchino@tugraz.at)

**Abstract.** Recently, seasonal changes in sea ice cover have been found to elevate basal melt rates of the Ross Ice Shelf (RIS) calving front at sensitive regions. Melting at these sensitive regions has been found to impact ice sheet mass balance. However, the influence of these seasonally elevated basal melt rates on RIS flow variability is not yet fully understood. This paper aims to explore whether seasonal perturbations in basal melt rates of the RIS can explain intra-annual variations in ice flow measured

by GNSS at four sites across the ice shelf. We use the automatic differentiation tool in the Ice-sheet and Sea-level System Model (ISSM) to identify regions of the RIS where changes in basal melt affect ice velocities at the GNSS sites. Next, we seasonally perturb Massachusetts Institute of Technology general circulation (MITgcm) basal melt rates in ISSM at these sensitive regions to try and replicate the GNSS ice flow observations. The GNSS observations display clear intra-annual velocity variability at the four sites, with two distinct peaks observed in austral summer and austral winter. We can replicate this intra-annual velocity

variation for GNSS sites near the calving front by seasonally perturbing the basal melt rates at the identified sensitive regions of the ice shelf. We argue that the perturbed seasonal basal melt variability at sensitive regions along the calving front is a realistic scenario for the RIS. Thus, we suggest that the GNSS-recorded intra-annual velocity variations along the calving front could be partly driven by seasonal changes in basal melting today. We also try to replicate intra-annual velocity variability observed at the Siple Coast by seasonally perturbing basal melt rates at sensitive regions there. However, we are unable to replicate similar

magnitudes of velocity variations to the GNSS measurements and suspect that the perturbed seasonal basal melt variability is unrealistic, with no observations of seasonally high basal melt rates at the Siple Coast grounding lines or pinning points. Thus, seasonal changes in basal melt cannot explain the observed intra-annual velocity variability at all the GNSS sites, and further work is needed. Our sensitivity maps highlight regions of the ice shelf where changes in basal melt most influence velocities, and are a valuable addition to fieldwork campaigns and modelling studies.



## 1 Introduction

The Antarctic Ice Sheet (AIS) contains the vast majority of Earth's freshwater and has the potential to raise global sea levels by 58 m (Mottram et al., 2019; Schlegel et al., 2018; Dirscherl et al., 2020). Over recent decades, the AIS has been losing mass at an accelerating rate due to the warming of the atmosphere and ocean (Pattyn et al., 2018; Shepherd et al., 2012, 2018; Jenkins et al., 2018; Rignot et al., 2019; Lipscomb et al., 2021). Ocean-forced basal melting and calving drive the largest mass losses on the AIS (Pattyn et al., 2018; Rignot et al., 2019; Adusumilli et al., 2020; Joughin et al., 2014). Floating ice shelves, in particular, provide buttressing to grounded ice and thus are vital for controlling AIS mass loss (Schoof, 2007; Gudmundsson, 2013; Dinniman et al., 2016; Joughin et al., 2013; Pattyn and Durand, 2013). Ocean-forced basal melting thins ice shelves, reducing their buttressing ability of the grounded ice, which in turn increases ice discharge and grounding line retreat (Depoorter et al., 2013; Joughin et al., 2013; Jenkins et al., 2018; Greene et al., 2022; Smith et al., 2020; Gudmundsson et al., 2019).

The Ross Ice Shelf (RIS) is Antarctica's largest ice shelf by area and is approximately in balance (Moholdt et al., 2014; Rignot et al., 2013; Depoorter et al., 2013) with average basal melt rates of approximately 0.1 m/a (Klein et al., 2020; Das et al., 2020). These basal melt rates are relatively low due to the cold dense water masses formed on the continental shelf blocking the sub-ice-shelf ocean cavity from warm Circumpolar Deep Water (CDW) intrusions (Moholdt et al., 2014; Stevens et al., 2020; Adusumilli et al., 2020). However, basal melt rates of the RIS vary spatially as they are driven by subsurface inflows of cold High Salinity Shelf Water (HSSW) near the grounding lines and seasonal inflows of summer-warmed Antarctic Surface Water (AASW) at the calving front (Stewart et al., 2019; Stevens et al., 2020; Klein et al., 2020; Jendersie et al., 2018; Dinniman et al., 2016; Adusumilli et al., 2020). Recently, high basal melt rates have been observed at the calving front near Ross Island due to the seasonal inflow of summer-warmed AASW from the adjacent Ross Sea Polynya downwelling into the ice shelf cavity (Stewart et al., 2019; Malyarenko et al., 2019). Ross Island has been identified as a sensitive region where changes in ice thickness can drive changes in ice shelf dynamics and mass balance (Reese et al., 2018; Gudmundsson et al., 2019; Fürst et al., 2016; Baldacchino et al., 2022). With predicted surface warming and declines in summer sea ice, these high basal melt rates along the calving front are projected to increase (Stewart et al., 2019; Dinniman et al., 2018; Schodlok et al., 2016; Smith Jr. et al., 2014). Additionally, high basal melt rates may occur in the future due to changes in the primary modes of basal melting of the RIS, such as increases in the amount of Modified Circumpolar Deep Water (mCDW) heat flux flowing onto the continental shelf and reducing the rate of sea ice formation (Tinto et al., 2019; Reddy et al., 2007). mCDW is formed by CDW flowing onto the continental shelf of the Ross Sea and mixing with the AASW and HSSW (Dinniman et al., 2018; Smith Jr. et al., 2014).

Understanding the flow variability of the RIS is essential for monitoring mass balance changes. Ice shelves flow by gravity-driven horizontal spreading with resistance to flow provided by shear at bay walls and pinning points (Hulbe et al., 2013; Cuffey and Paterson, 2010). Changes in ice shelf dynamics, geometry, and mass can lead to changes in velocity on floating and grounded ice (Fürst et al., 2016; Reese et al., 2018; Gudmundsson et al., 2019; Mosbeux et al., 2023). The RIS has typical flow speeds of several hundred meters per year, with the active Siple Coast Ice Streams and Byrd Glacier displaying velocities of >



300 m/a (Figure 1). The ice shelf front exhibits the fastest flow rates of 800 - 1200 m/a (Figure 1) (Rignot et al., 2017). The

Siple Coast Ice Streams and Transantarctic Mountain outlet glaciers are the main conduits of ice discharging into the RIS from the West Antarctic Ice Sheet (WAIS) and East Antarctic Ice Sheet (EAIS), respectively (Shabtaie and Bentley, 1987; Bennett, 2003; Bindschadler et al., 2003; Catania et al., 2012b) (Figure 1). The Siple Coast Ice Streams are underlain by deformable till that is water-saturated (Kamb, 1991; MacAyeal, 1992; Joughin et al., 2004) and basal friction at their beds and along the lateral margins are the primary forces resisting flow (Ranganathan et al., 2021). The Kamb Ice Stream (KIS) has been inactive

for the last 160 years (Retzlaff and Bentley, 1993; Thomas et al., 2013), and the Whillans and Mercer ice streams are slowing down and could reach stagnation in the next 50 years (Thomas et al., 2013). In the Transantarctic Mountains, flow of the Byrd Glacier is driven by high driving stresses modulated by changes in subglacial hydrology (Whillans et al., 1989; Van Der Veen et al., 2014; Stearns. et al., 2008).

Currently, the influence of seasonal oceanic and atmospheric variability on the flow dynamics of the RIS is unknown.

Previous studies have suggested that RIS velocities may be modulated at seasonal to interannual timescales by basal melting at the calving front (Stewart et al., 2019; Tinto et al., 2019). It is important to understand the impact of this seasonality on ice shelf flow as this may provide further understanding of the processes controlling ice shelf mass balance over longer timescales (Mosbeux et al., 2023; Dutrieux et al., 2014; Paolo et al., 2018; Jenkins et al., 2018). Global Navigation Satellite System (GNSS) receivers near-continuously record at high temporal resolution throughout the winter and summer seasons and thus

have the unique ability to measure seasonal variations in ice velocities (Brunt, 2008; King et al., 2011; Brunt and Macayeal, 2014). Typically, GNSS receivers are employed to measure ice velocities over 1–3 months in the austral summer and to highlight short timescale processes such as tidal fluctuations within one season. However, previous studies have observed inter-annual (monthly to seasonal) velocity variations using GNSS instruments deployed on the RIS over multiple seasons (Klein et al., 2020; Mosbeux et al., 2023). Klein et al. (2020) use ice sheet modelling to conclude that these inter-annual velocity

variations are not driven by seasonal basal melt rates and that some other forcing must be driving the velocity variations for one site on the RIS. More recently, Mosbeux et al. (2023) used a viscoelastic model, concluding that the observed intra-annual velocity variations on the RIS are driven by seasonal variations in Sea Surface Height (SSH). Mosbeux et al. (2023) were able to reproduce the GNSS measured intra-annual velocity variability if a sufficiently large cycle of SSH-induced basal shear stress change near the grounding line was parameterized in their ice sheet model (Mosbeux et al., 2023).

This paper aims to explore whether perturbations in basal melt rates on the RIS can explain the observed intra-annual variations in ice velocities at four sites across the RIS. This paper presents three new long-duration (12 - 24 months) GNSS measurements of intra-annual ice velocity variations on the RIS. We explore four sites: the shear margin near Ross Island (Site 1), the calving front near Coulman High (Site 2), a mid-shelf region (Site 3), and the KIS grounding zone (Site 4; Figure 1). The GNSS receivers display clear intra-annual velocity variability, with two distinct velocity peaks each year: one in the

austral summer and one in austral winter. Baldacchino et al. (2022) showed that RIS mass balance is sensitive to changes in basal melt at specific locations. We apply here a similar approach to identify the regions where flow speed at GNSS locations is most sensitive to melt, and we further explore if changes in basal melt at these locations can explain the observed velocity time series.





## 2    Locations and Methods

### 2.1    Global Navigation Satellite Systems locations

Figure 1 highlights the locations of the four Ross Ice Shelf GNSS sites explored in this paper. Firstly, GNSS Site 1 is located close to Ross Island, which is a major pinning point making it a sensitive region where changes in ice thickness influence the flow speed and mass balance of the entire ice shelf (Gudmundsson et al., 2019; Fürst et al., 2016; Baldacchino et al., 2022; Reese et al., 2018) (Figure 1). Pinning points provide resistance to ice shelf flow by modifying the balance of forces within the floating ice (Still et al., 2019; Cuffey and Paterson, 2010). This modification of forces has an effect everywhere on the ice shelf due to the balance of forces in floating ice being non-local (Still et al., 2019; Cuffey and Paterson, 2010). High basal melt rates have been observed close to the Ross Island pinning point (Stewart et al., 2019). These basal melt rates may thin the ice, reduce the resistance of the pinning point and drive changes in ice shelf flow.

Site 2 is located at Coulman High close to the calving front and is likely to be influenced by seasonal changes in basal melting (Figure 1). High basal melt rates have been observed in this region due to declines in sea ice cover and warming of the AASW during the austral summer (Stewart et al., 2019). Site 2 is located within the "passive" region of the ice shelf and thus this region can be removed without impacting the mass balance of the ice shelf (Fürst et al., 2016). However, thinning of this region or changes in the ice-front location (i.e., via iceberg calving) will alter the stress balance and velocities of the ice shelf (Gudmundsson et al., 2019; Klein et al., 2020).

Site 3 is located in the mid-shelf region of the RIS (200km from the calving front) and is the same site (referred to as DR10) also reported in Klein et al. (2020) and Mosbeux et al. (2023) (Figure 1). Ice flow in the central portion of the RIS is primarily extensional which leads to along-flow thinning (Das et al., 2020). There are no pinning points or ice rises within the vicinity of Site 3, and no observations of high basal or surface melt rates here.

Finally, Site 4 is located at the Kamb Ice Stream (KIS) grounding line (Figure 1). The KIS has been inactive for the last 160 years likely due to a change in subglacial hydrology (Retzlaff and Bentley, 1993; Thomas et al., 2013; Hulbe et al., 2016). The KIS used to flow at speeds of 350 m/a but presently flows at speeds of less than 5 m/a (Rignot et al., 2017). Studies have indicated that the KIS could reactivate this century due to its hydrological setting and the length of time it has been inactive (Bougamont et al., 2015; van der Wel et al., 2013).

### 2.2    Global Navigation Satellite Systems data and processing

Three of the GNSS units were installed during field seasons 2019/2020 and data was downloaded in December 2021 (Sites 1, 2, and 4 in Figure 1). These GNSS units were battery-powered and deployed on the RIS for multiple field seasons to provide a long-term continuous dataset that can observe intra-annual velocity variations. Site 1 GNSS unit recorded every 30 seconds for 1 hour every 6 hours, while the Site 2 GNSS unit recorded every 30 seconds over 24 hours. Site 4 GNSS unit recorded every 30 seconds and operated continuously, but was shifted approximately 2.7 km upstream in December 2020. The GNSS unit at Site 3 is the same as that in Klein et al. (2020) and is described in more detail in their paper (referred to as DR10) (Figure 1). Site 1 GNSS unit has 80 days of data missing in July - October 2020, and 70 days in July - September 2021, while Site 2 GNSS



unit has 104 days of data missing in June - November 2020 and 30 days in July - August 2021. The Site 3 GNSS unit recorded every 30 seconds over 24 hours for 1 year (2015 - 2016), with a few days dropped in the austral winter of 2016 (Klein et al., 2020).

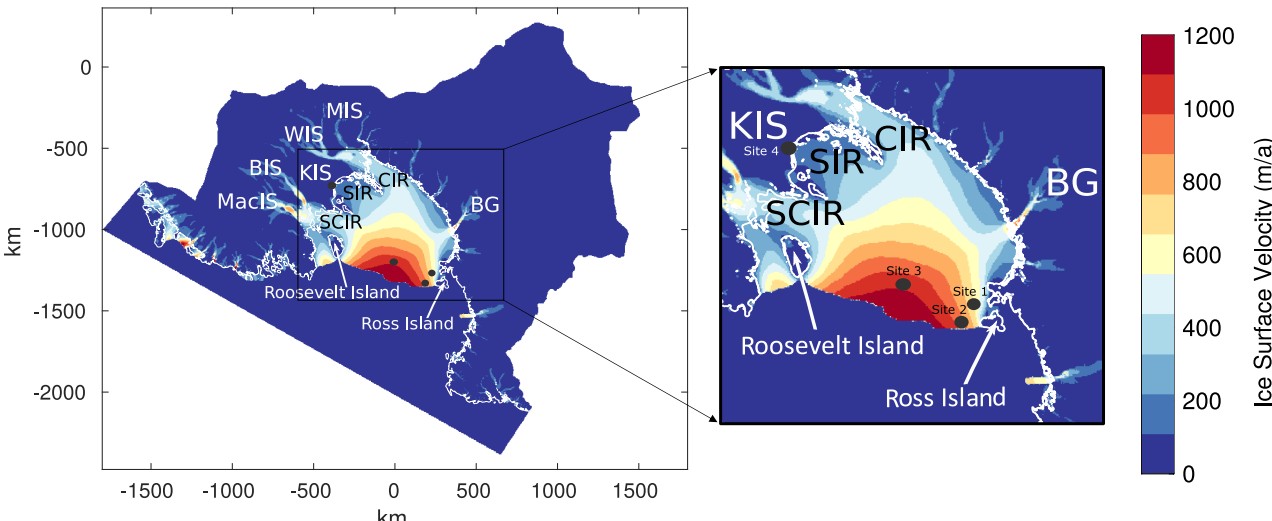

**Figure 1.** Modelled Ross Ice Shelf surface velocities after initialisation. The grounding line is marked in white. The GNSS sites are shown: Site 1 (shear margin region), Site 2 (Coulman High region), Site 3 (mid-shelf region), and Site 4 (KIS grounding zone). Locations discussed in this paper are also labelled. These include the Siple Coast Ice Streams: Mercer Ice Stream (MIS), Whillans Ice Stream (WIS), Kamb Ice Stream (KIS), Bindschadler Ice Stream (BIS), and MacAyeal Ice Stream (MacIS). Byrd Glacier (BG) and Ross Island are also labelled. In addition, the ice rises are labelled on the Siple Coast: CIR = Crary Ice Rise, SIR = Steershead Ice Rise, SCIR = Shirase Coast Ice Rumples, and Roosevelt Island. The projection of this map and all others presented is polar stereographic with a true scale at -71° (EPSG:3031).

GNSS data were processed using the Precise Point Positioning (PPP) methodology (Zumberge et al., 1997) and Natural Resources Canada's Canadian Spatial Reference System Precise Point Positioning (CSRS-PPP) post-processing service[1]. For the 30-second sampled continuous data (sites 2, 3, and 4), data were divided into 3-hour segments and processed statically to obtain a single position every 3 hours. For Site 1, which has a different sampling frequency, the data were divided into 1-hour segments every 6 hours and a single position was obtained every 6 hours. Data processing was iterated whereby the initial

positions were updated with the first processing results and then reprocessed to obtain new position solutions. The position solutions were projected into polarstereographic coordinates (EPSG:3031) and then used to estimate site velocity by weighted linear regression through x and y coordinates. The position weightings were provided by the reported processing uncertainty. Regression gradients provided velocities in the x and y direction ($v_x$, $v_y$) with gradient uncertainties propagated to provide uncertainties in velocity and direction. The linear regression of positions was estimated at every time step (either 3 hourly or

6 hourly) over centered time windows of 8 weeks duration. This provides a low-noise time series with a high spatial fidelity

---

[1]https://webapp.csrs-scrs.nrcan-rncan.gc.ca/geod/tools-outils. Last accessed: 15.08.2023





(albeit smoothed) that shows the seasonal cycle in velocity without aliasing spring-neap tidal velocity signals. The use of the 8-week window to estimate velocity means a stepwise increase in velocity would appear as a smooth increase in velocity beginning 4-weeks earlier. Other time window lengths were tested and the seasonal signal was seen to be largely independent of the length used. The resulting uncertainties were low with 99% of the $1\sigma$ velocity uncertainties less than 0.04 m/a for sites 1 and 3, less than 0.06 m/a for Site 2, and less than 0.01 m/a for Site 4. We present all velocities as the deviation from the initial velocity to facilitate comparison with the modelling results.

## 2.3 Automatic Differentiation

We used Automatic Differentiation (AD, Sagebaum et al., 2019) in the Ice-sheet and Sea-level System Model (ISSM) to explore the influence that changes in basal melt have on the velocity at each GNSS site. The complete model description is available in Baldacchino et al. (2022). Here, instead of computing the sensitivity of the model's final volume above flotation, we were interested in the sensitivity of the model velocity at these four GNSS sites. AD allowed us to efficiently map by how much the velocity at each site would be affected if we perturbed the ocean-induced melt at the scale of the model mesh.

The model domain covered the entire RIS and has a non-uniform mesh with a resolution of 1 km at the grounding lines and in the shear margins, 20 km in the ice sheet interior, and at most 10 km within the ice shelf. The basal friction co-efficient over grounded ice and the ice viscosity parameter of the floating ice, $B$, was inferred through a data assimilation technique (Morlighem et al., 2010, 2013) to reproduce observed InSAR surface velocities from the MEaSURES data-set (Rignot et al., 2017; Baldacchino et al., 2022). Environmental boundary conditions included RACMO2.3p2 Surface Mass Balance (Van Wessem et al., 2018) and Massachusetts Institute of Technology general circulation (MITgcm) basal melt rates (Losch, 2008; Holland and Jenkins, 1999; Davis and Nicholls, 2019; Baldacchino et al., 2022). The ice sheet model was run forward for 20 years to allow the grounding line position and ice geometry to relax.

After relaxation, we ran the AD model for 6 months and evaluated the sensitivity of the final velocity at each of the four GNSS sites to perturbations in melting rates under floating ice, $\dot{M}_b$. Automatic differentiation provided the gradient of the final velocity at each site, $v_i$, to basal melt: $\mathcal{D}v_i(\dot{M}_b)$. In other words, the first order response of the velocity to a given perturbation $\epsilon\delta\dot{M}_b$ in $\dot{M}_b$ (where $\epsilon \in \mathbb{R}$, and $\delta\dot{M}_b$ was defined over the entire model domain $\Omega$ that can be spatially variable) was given by:

$$v_i(\dot{M}_b + \epsilon\delta\dot{M}_b) = v_i(\dot{M}_b) + \epsilon \int_{\Omega} \mathcal{D}v_i(\dot{M}_b)\, \delta\dot{M}_b\, d\Omega + \mathcal{O}\left(\epsilon^2\right). \tag{1}$$

The gradient, $\mathcal{D}v_i(\dot{M}_b)$ (in m$^{-2}$), therefore highlighted the regions where the modelled velocity at a given site was most sensitive to changes in $\dot{M}_b$, and the regions where changes in $\dot{M}_b$ would not affect the final velocity at a first order.

This approach provided four sensitivity maps, one for each site. Figure 2 shows the areas where this sensitivity was higher than our threshold value of 2e-11 m$^{-2}$. The sensitivity value of 2e-11 m$^{-2}$ was chosen arbitrarily, with these areas deemed highly sensitive to basal melt changes. Choosing a lower sensitivity threshold would enlarge the surface area over which the perturbation would need to be applied, and a higher sensitivity threshold would have the opposite effect. We chose a sensitivity value of 2e-11 m$^{-2}$ to highlight areas of high sensitivity over a surface area that is not too restrictive or extensive across the ice shelf. We also included a lower sensitivity value of 0.5e-11 m$^{-2}$ (Figure A1) in our experiments to highlight that the modelled





velocity variations are similar for both sensitivity thresholds. These regions, highlighted in dark red, show where an increase
in basal melt rates leads to an increase in ice velocity for each site, and therefore where changes in melt rates would impact ice
velocity at these sites the most. We can see that these sensitive regions are sometimes hundreds of kilometres from the GNSS
stations.

## 2.4 Modelled perturbed basal melt

A set of modelling experiments within ISSM was then performed, where the MITgcm basal melt rates were perturbed season-
ally (using a simple sine function) at regions identified as sensitive in the final AD map (red regions in Figure 2):

$$
\dot{M}_b(t) = \begin{cases} \text{MITgcm}(t) + p \sin\left(4\pi + 3t\right), & \text{if one or more maps shows a sensitivity} > \text{2e-11 m}^{-2} \\ \text{MITgcm}(t), & \text{otherwise} \end{cases}
\tag{2}
$$

where MITgcm$(t)$ was the unperturbed melt rate from the MITgcm, and $p$ was the amplitude of the perturbation, taken here
as 0, 20, 40, 60 or 80 m/a. A sine function with two peaks was used to simulate two basal melt peaks per year (these basal
melt peaks were either 20, 40, 60 or 80 m/a depending on the perturbation). These peaks were modelled in April and October
using the + 3 radiant in the sine function. These two basal melt peaks per year were needed to replicate the observed intra-
annual velocities at the GNSS sites. The raw MITgcm basal melt rates displayed a seasonal signal, however, the amplitude of
this seasonal variability was not large enough and the phasing incorrect to replicate the GNSS observed velocity variability.
Therefore, we simulated a set of perturbations on top of the raw MITgcm basal melt rates, as described in equation (2). This
model was run forward for an additional 20 years to allow the geometry and grounding line to stabilize. The model was then
run forward 2 years using the same model setup as described above.

## 3 Results

### 3.1 GNSS Velocities

#### 3.1.1 Site 1

GNSS velocity observations for Site 1 are presented in Figure 3. Site 1's velocities range from a maximum of 447 m/a to a
minimum of 441 m/a with a clear decrease in velocities of 4 m/a over the two years (Figure 3). The velocity variations are
small throughout the two years, however, an intra-annual signal is observed at Site 1 (Figure 3). Figure 3 displays two velocity
peaks: one in June (austral winter) and one in January (austral summer). These velocity peaks are preceded by periods of
acceleration (April - June and November - January) and periods of deceleration (February - April and July - October) (Figure
3). An acceleration of 2 m/a for the velocity peak in June 2020, 1.5 m/a for the velocity peak in January 2021 and 1.5 m/a for
the velocity peak in June 2021 highlights the largest seasonal velocity variations at Site 1 (Figure 3).





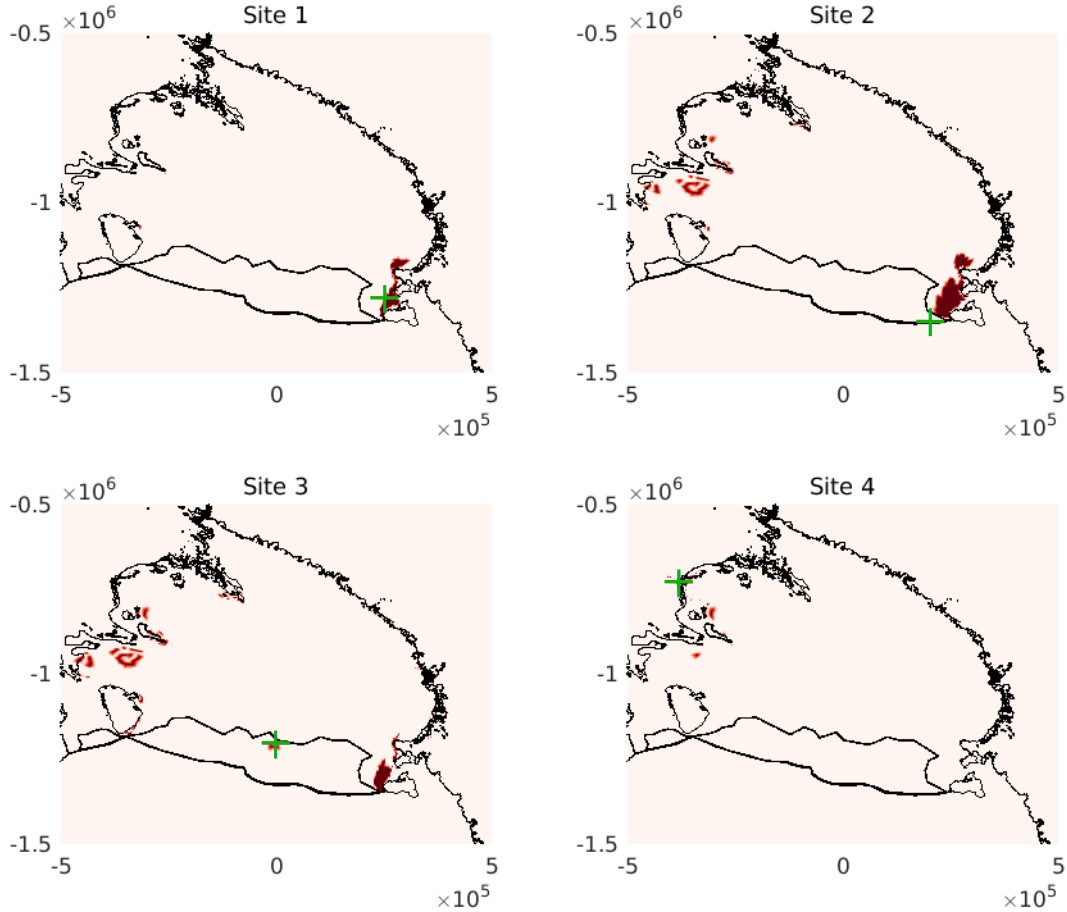

**Figure 2.** Locations where the basal melt rate was perturbed seasonally (i.e., where the AD mapped sensitivity is greater than $2e-11 \text{ m}^{-2}$ (dark red)) for each GNSS site (green markers). The passive ice on the RIS identified by Fürst et al. (2016) is also outlined in black.

### 3.1.2 Site 2

GNSS velocity observations for Site 2 are presented in Figure 3. The velocities range from a maximum of 745 m/a to a minimum of 739 m/a, with a clear intra-annual signal observed at Site 2 (Figure 3). Two distinct velocity peaks are observed at Site 2: one in December (austral summer) and one in July (austral winter). These velocity peaks are preceded by periods of acceleration (April - July and October - December) and periods of deceleration (January - April and July - October) (Figure 3). Similar to Site 1, the velocity variations are small throughout the two years, however, a larger acceleration is observed in the lead-up to the velocity peaks compared to Site 1. An acceleration of 5 m/a for the velocity peak in July 2020, 1.5 m/a for the velocity peak in December 2020 and 3 m/a for the velocity peak in July 2021 highlights the largest seasonal velocity variations

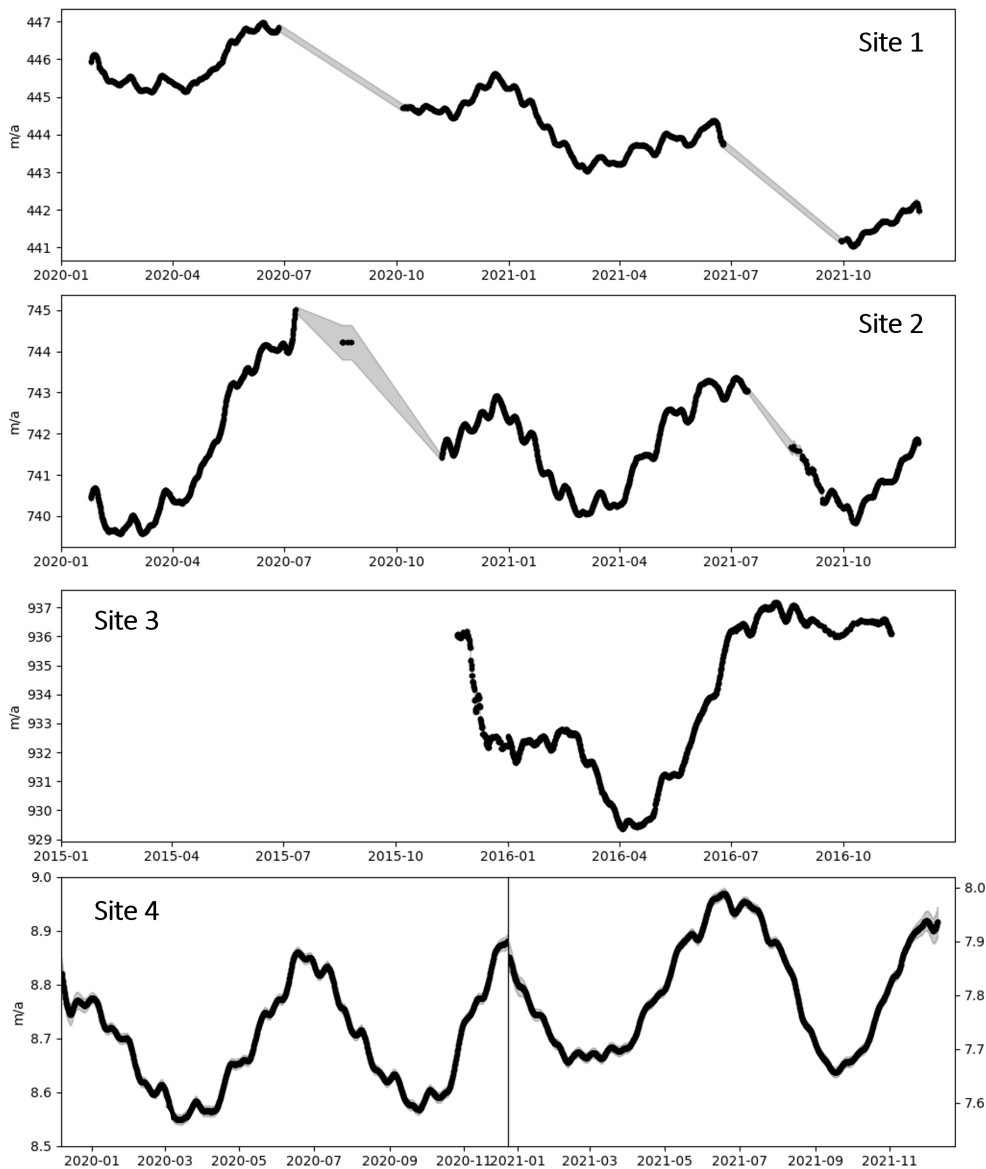

**Figure 3.** The GNSS velocities (in m/a) at Site 1 (shear margin region close to Ross Island), Site 2 (Coulman High region close to the calving front), Site 3 (mid-shelf region) and Site 4 (Kamb Ice Stream grounding line). The errors are provided in the grey windows enclosing the black lines. These errors are not visible in a few places, as the errors are very small.

at Site 2 (Figure 3). Site 2 displays a larger maximum velocity of 745 m/a compared to Site 1's maximum velocity of 447 m/a.
Overall, the velocities neither decreased nor increased significantly throughout the two years at Site 2, with the variations in velocities reflecting a clear intra-annual signal.





### 3.1.3  Site 3

GNSS velocity observations for Site 3 are presented in Figure 3. The velocities range from a maximum of 937 m/a to a minimum of 929 m/a and thus display higher maximum velocities compared to Sites 1 and 2. However, Site 3's intra-annual
signal is not similar to Sites 1 and 2, with a smaller velocity peak observed in March (austral summer) and a larger velocity peak in August (austral winter) (Figure 3). These velocity peaks are preceded by periods of acceleration (January - March and April - August) and periods of deceleration (March - April and September - December) (Figure 3). Similar to Sites 1 and 2, the velocity variations throughout the year are small, however, Site 3 displays the largest acceleration in the lead-up to a velocity peak. A small acceleration of 1 m/a for the velocity peak in March 2016 and a much larger acceleration of 8 m/a for the velocity
peak in August 2016 is observed (Figure 3).

### 3.1.4  Site 4

GNSS velocity observations for Site 4 are presented in Figure 3. Figure 3 shows that Site 4 has the smallest maximum velocities of 9.0 m/a compared to the other GNSS sites. The KIS has been inactive for the last 160 years and thus the velocities are very low at the grounding line compared to Sites 1, 2, and 3. Site 4 displays a clear intra-annual signal which is similar to Sites 1
and 2 (Figure 3). Two velocity peaks are observed at Site 4 for the years 2020 and 2021: one in December (austral summer) and one in June (austral winter) (Figure 3). Site 4 has the most complete record of GNSS velocity measurements for two years and thus highlights the intra-annual velocity variation nicely. These velocity peaks are preceded by periods of acceleration (March - June and October - December) and periods of deceleration (January - March and July - August) (Figure 3). Site 4 displays the smallest velocity variations compared to the other GNSS sites with an acceleration of 0.4 m/a for the velocity peak in June
2020, 0.3 m/a for the velocity peak in December 2020, 0.3 m/a for the velocity peak in July 2021 and 0.3 m/a for the velocity peak in December 2021 (Figure 3). Klein et al. (2020) also observed that the further the GNSS sites are from the calving front, the smaller the intra-annual velocity variation.

A fortnightly signal is found in the displacement at all GNSS sites and we attribute this to the response of the ice shelf to spring-neap variability in the tidal cycle (Padman et al., 2003; Ray et al., 2021; Rosier and Gudmundsson, 2020). This
fortnightly tide-forced variability is dampened by our use of an 8-week window for our velocity estimates (Mosbeux et al., 2023).

### 3.2  Sensitivity Maps

The AD model produced sensitivity maps show that high sensitivity is observed at the pinning points and ice rises downstream of the Siple Coast ice streams (i.e., Roosevelt Island, Crary Ice Rise, Steershead Ice Rise, and the Shirase Coast Ice Rumples)
for all GNSS sites (Figure 4. For GNSS Sites 1, 2, and 3 we also see high sensitivity to changes in basal melting at the calving front near the Ross Island pinning point. Changes in basal melting can result in detachment from pinning points and ice rises resulting in changes in ice speed (Still et al., 2019; Baldacchino et al., 2022; Reese et al., 2018). Ross Island is a structurally critical region and Gudmundsson et al. (2019) found that rapid melting there influences the flow speed of the entire RIS. Our





sensitivity maps confirm this finding, highlighting that changes at and/or near the Ross Island pinning point influence velocities

at Sites 1, 2, and 3. It is also important to highlight that Sites 1 and 2 are situated close to the Ross Island pinning point, and thus have high sensitivity to local changes in basal melt.

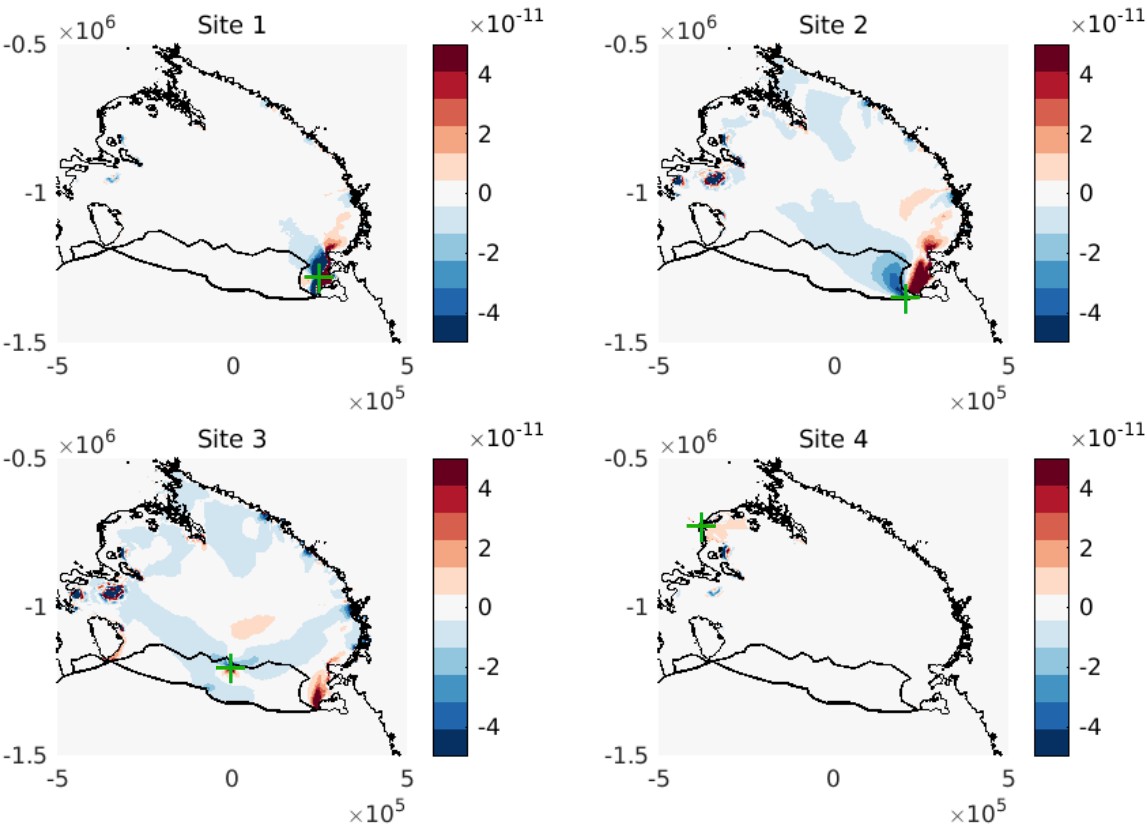

**Figure 4.** Sensitivity maps of the final velocity at each of the four GNSS sites to basal melting rates under floating ice $\dot{M}_b$ over 40 years (in $m^{-2}$). The sensitivity maps highlight that an increase or decrease in basal melt rates at identified sensitive regions increases or decreases the velocities at the GNSS sites. The passive ice on the RIS identified by Fürst et al. (2016) is outlined in black. The GNSS sites are identified using green markers.

Additionally, high sensitivity is observed at the Siple Coast Ice Streams and Byrd Glacier grounding lines for Sites 2 and 3 (Figure 4). The grounding lines show high sensitivity because changes in basal melting there can lead to changes in basal friction and grounding line retreat (Baldacchino et al., 2022). These changes in basal friction can drive changes in the ice

streams and outlet glaciers' flow dynamics and discharge (Baldacchino et al., 2022; Pattyn, 2017; Shepherd et al., 2018). We observe high sensitivity at the near-stagnant KIS grounding zone for Site 4, and no sensitivity elsewhere for this GNSS site. This high sensitivity at the KIS grounding zone highlights that local changes in basal melt influence the velocities at Site 4 and changes in basal melt elsewhere on the ice shelf do not affect Site 4 velocities.



Finally, high sensitivity within the interior of the ice shelf and directly downstream of active ice streams and outlet glaciers is observed for GNSS Sites 2 and 3 (Figure 4). Sensitivity to changes in basal melting is also observed at the "passive" region (black outline in Figure 4 identified by Fürst et al. (2016)) for GNSS Sites 2 and 3. This indicates that local changes in basal melt affect the velocities at Sites 2 and 3 as both these sites are located in the "passive" region. Overall, the sensitivity maps show that GNSS Sites 2 and 3 velocities have high sensitivity to basal melting across the majority of the ice shelf, compared to Sites 1 and 4, which have higher sensitivities to local changes in basal melting.

## 3.3 Modelled Velocities

The modelled velocity variations are compared to the GNSS velocity variations (change from the initial velocity) for each site in Figure 5. We model two distinct velocity peaks: one in January (austral summer) and one in June (austral winter). We perturbed the basal melt rates seasonally at the identified sensitive regions to replicate these two velocity peaks at all the GNSS sites. These sensitive regions are located at the calving front near the Ross Island shear zone for Sites 1, 2 and 3 and pinning points downstream of the Siple Coast Ice Streams and at their grounding lines for Sites 2, 3 and 4. For all GNSS sites, we observe that the intra-annual velocity variation is small when we perturb the basal melt rates by a magnitude of 20 m/a and this intra-annual velocity variation quadruples when we perturb the basal melt rates by a magnitude of 80 m/a (Figure 5). The dotted black line in Figure 5 highlights the model experiments which used the lower sensitivity value of 0.5e-11 $m^{-2}$ (Figure A1). Figure 5 shows that for all GNSS sites the use of the lower sensitivity value did not significantly affect the final modelled velocity variations. The modelled absolute velocity values are similar to the GNSS absolute velocity values for Site 1 but are larger than the GNSS absolute velocity values for Sites 2, 3 and 4. However, in this paper, we focus on replicating the GNSS velocity variations in our model simulations and thus, focus on these results.

Our model predicts an intra-annual variation in velocity at Site 1 ranging from 1 m/a to 5 m/a for 20 m/a basal melt perturbation, 4 m/a to 12 m/a for 40 m/a basal melt perturbation, 5 m/a to 18 m/a for 60 m/a basal melt perturbation and 6 m/a to 28 m/a for the 80 m/a basal melt perturbation (Figure 5). An acceleration of 4 m/a for the velocity peaks in January and June is observed in the 20 m/a basal melt perturbed model experiments, which is most similar to Site 1's GNSS observed accelerations of 2 m/a for the velocity peak in June 2020, 1.5 m/a for the velocity peak in January 2020 and 1.5 m/a for the velocity peak in June 2021 (Figure 5).

The modelled intra-annual velocity variations at Site 2 range from 0 m/a to 3 m/a for 20 m/a basal melt perturbation, 2 m/a to 6 m/a for 40 m/a basal melt perturbation, 2 m/a to 9 m/a for 60 m/a basal melt perturbation and 2 m/a to 13 m/a for the 80 m/a basal melt perturbation (Figure 5). The phasing of the modelled velocity peaks (January and June) are offset by one month compared to the GNSS observed velocity peaks (December and July) at Site 2 (Figure 5). However, the peak in velocity occurs at the end of December, and the beginning of July which is similar to our modelled velocity peaks in January and June. The 20 m/a basal melt perturbed modelled velocity variation is similar in amplitude to the GNSS velocity variations. An acceleration of 3 m/a for the velocity peaks in January and June is observed in the 20 m/a basal melt perturbed model experiments, which is most similar to Site 2's GNSS observed accelerations of 5 m/a for the velocity peak in July 2020, 1.5 m/a for the velocity peak in December 2020 and 3 m/a for the velocity peak in July 2021 (Figure 5).





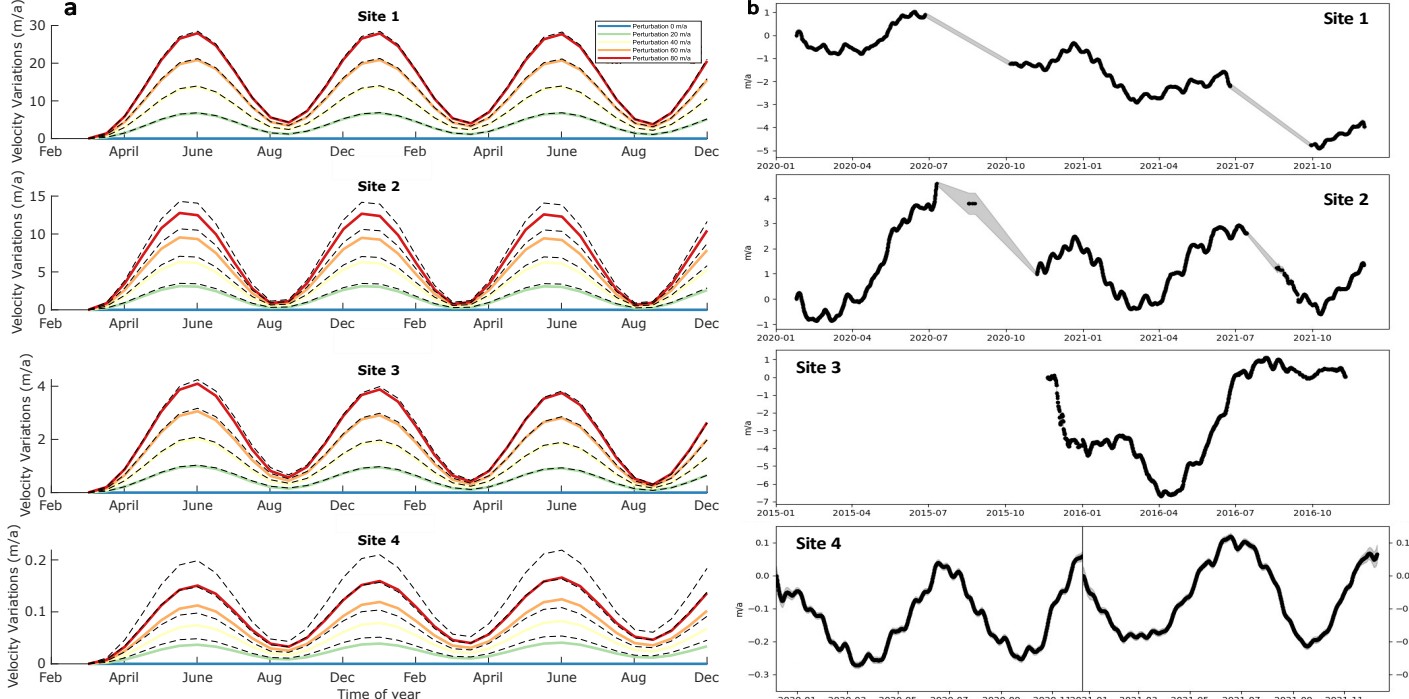

**Figure 5.** The GNSS (right) and modelled (left) velocity variations (in m/a) at each GNSS site: Site 1 (shear margin region), Site 2 (Coulman High region), Site 3 (mid-shelf region), and Site 4 (KIS grounding zone). The dotted black line represents the additional sensitivity threshold value experiment.

For Site 3, the modelled intra-annual velocity variations range from 0 m/a to 1 m/a for 20 m/a basal melt perturbation, 0.5 m/a to 2 m/a for 40 m/a basal melt perturbation, 1 m/a to 2.5 m/a for 60 m/a basal melt perturbation and 1.5 m/a to 4 m/a for

the 80 m/a basal melt perturbation (Figure 5). The modelled velocity peaks occur in January and June, which is different from the GNSS observed velocity peaks in March and August for Site 3 (Figure 5). Therefore, the phasing of the modelled velocity variations is offset by a couple of months. Additionally, the amplitude of the modelled velocity peaks is smaller compared to the GNSS observed velocity peaks at Site 3. An acceleration of 1 m/a for the velocity peak in March 2016 and 8 m/a for the velocity peak in August 2016 is observed by the GNSS receiver at Site 3 (Figure 5). None of the basal melt perturbed

modelled velocity variations capture an acceleration of 8 m/a in August (Figure 5). Overall, the amplitude of the modelled velocity variation is significantly smaller compared to the GNSS measurements at Site 3.

Finally, we model the smallest intra-annual velocity variation at Site 4, which is similar to the GNSS measurements (Figure 5). The modelled intra-annual velocity variations at Site 4 range from 0.01 m/a to 0.04 m/a for 20 m/a basal melt perturbation, 0.02 m/a to 0.06 m/a for 40 m/a basal melt perturbation, 0.03 m/a to 0.1 m/a for 60 m/a basal melt perturbation and 0.04 m/a

to 0.15 m/a for the 80 m/a basal melt perturbation (Figure 5). The phasing of the modelled velocity variations is similar to the GNSS-measured velocity variations with a clear intra-annual signal observed at Site 4. The modelled velocity peaks occur



in January and June which is similar to the GNSS-measured velocity peaks at the end of December and in June (Figure 5). However, none of the basal melt perturbed modelled velocity variations could replicate the amplitudes of the GNSS observed velocity variations. An acceleration of 0.4 m/a for the velocity peak in June 2020, 0.3 m/a for the velocity peak in December 2020, 0.3 m/a for the velocity peak in July 2021 and 0.3 m/a for the velocity peak in December 2021 (Figure 5). An acceleration of 0.11 m/a for the velocity peaks in January and June is observed in the 80 m/a basal melt perturbed model experiments and is most similar to the GNSS-measured velocity variations at Site 4. Similarly, to Site 3, overall the amplitude of the modelled velocity variation is significantly smaller compared to the GNSS measurements at Site 4.

For all GNSS sites, an increase in basal melt rates accelerates the velocities compared to the control run (Figure 5). We would expect this as the seasonally elevated melt rates lead to short-term thinning and acceleration of the ice shelf (Gudmundsson et al., 2019; Campbell et al., 2018). The amplitude of modelled velocity variability decreases with increasing distance from the calving front, with the largest velocity variations modelled at Sites 1 and 2, and the smallest at Site 4. However, the GNSS units record the largest amplitude in velocity variation at Site 3 which also agrees with Klein et al. (2020) and Mosbeux et al. (2023).

## 4    Discussion

### 4.1    Intra-annual velocity variability

There is a clear seasonal signal for all GNSS sites with two velocity peaks: one in austral summer and one in austral winter (Figures 3 and 5). We observe that the magnitude of variation in velocities is larger for Sites 1, 2, and 3 compared to Site 4, likely due to the proximity of these sites to the calving front and seasonally elevated basal melt rates (Klein et al., 2020; Mosbeux et al., 2023). We model a similar intra-annual velocity variability (in both phasing and magnitude) at GNSS Sites 1 and 2 using seasonally perturbed basal melt magnitudes of 20 m/a at the identified sensitive regions (Figures 2, 4 and 5). However, for Site 3, the magnitude of the modelled velocity variability is significantly smaller and the phasing of the modelled velocity variability is offset by a couple of months compared to the GNSS measurements for all basal melt perturbations (Figure 5). Additionally, for Site 4 the modelled velocity variability has a similar phasing to the GNSS measurements, but an 80 m/a basal melt perturbation is needed at this site to model a similar magnitude in velocity variability. This may indicate that larger basal melt rates are needed at sensitive regions to replicate the intra-annual velocity variations at Sites 3 and 4 or a combination of mechanisms may be driving these velocity variations. The ice sheet model used in this paper does not account for varying forcings other than the seasonal cycle of basal melting.

In the modelling experiments, basal melt rates that peak in October and April are required to replicate the observed velocity peaks in austral summer (December and/or January) and austral winter (June and/or July) at Sites 1, 2 and 4 (Figures 3 and 5). Our modelled basal melt perturbations highlight that there is a delay of 2 - 3 months between the peak basal melt rates and peak velocities. We suggest this delay is due to the time it takes for the basal melt rates to thin the ice shelf significantly enough to change the stress regime at the GNSS sites. Previous studies have found that lag times in the ice shelf flow response occur due to the time required for basal melting to cause sufficient thinning (thickening) to produce an observable acceleration





(deceleration) of the ice shelf (Christianson et al., 2016; Greene et al., 2017; Roberts et al., 2018). Roberts et al. (2018) suggest that the time delay between basal melting and velocity change depends on the proximity of the region of interest to the highest basal melt rates and the thickness of that region (i.e., thicker areas have a greater lag time) (Greene et al., 2018). In this paper, we are perturbing basal melt rates at identified sensitive regions of the ice shelf that exist both locally and up to 1000km away from the GNSS sites (Figure 2). We hypothesize that the velocities reach a maximum in austral summer (December and/or

January) and austral winter (July and/or June) as the ice within these sensitive regions has thinned significantly during periods of peak melting in October and April (Figures 3 and 5). Greene et al. (2018) also found that the Totten Ice Shelf velocity maxima occurs at the end of the high melt season in July when the ice thickness reaches a minimum.

Sensitive regions identified from the AD experiments are located often at or near pinning points (Figure 2). Such areas are important for ice shelf stability and buttressing (Pattyn, 2017; Still et al., 2019; Matsuoka et al., 2015; Gudmundsson et al.,

2019; Lhermitte et al., 2020). Figure 2 highlights that basal melt rates are perturbed at the Ross Island region for Sites 1, 2 and 3. Ross Island is an important pinning point for the RIS, with changes in ice thickness here found to significantly impact overall ice shelf dynamics (Reese et al., 2018; Gudmundsson et al., 2019). On floating ice shelves, basal drag is negligible and the lateral drag depends on the existence of ice shelf margins and/or pinning points (Dupond and Alley, 2005; Gudmundsson et al., 2019; Gudmundsson, 2013). Ice shelf thinning can reduce the thickness of shear margins and the contact area over pinning

points and ice rises, which decreases the lateral drag and the buttressing ability of the ice shelf (Shepherd et al., 2004; Joughin et al., 2021; Gagliardini et al., 2010; Feldmann et al., 2022). This reduction in buttressing has a near-instantaneous effect on ice shelf acceleration (Larter, 2022; Arndt et al., 2018; Joughin et al., 2021; Dupond and Alley, 2005; Gudmundsson et al., 2019). Roberts et al. (2018) also observed that ice shelf thinning (thickening) was coincident with faster (slower) velocities due to changes in resistive stresses at ice rumples present on the Totten Ice Shelf. Therefore, thinning at and/or near pinning points

and ice rises will reduce the resistive stresses on the ice shelf and impact the velocities at the GNSS sites. The observed lag is likely due to the time it takes for the basal melt rates to thin the ice shelf enough at the identified sensitive regions to reduce the resistive stresses.

Additionally, we perturb basal melt rates at the KIS grounding zone to try and replicate the intra-annual signal observed at Site 4. Changes in basal melting at grounding zones will dictate the temperature gradient of the ice and the rate of basal

melting, affecting basal friction and ice rigidity around the grounding line (Baldacchino et al., 2022; Ranganathan et al., 2021). This can cause changes in basal stresses, grounding line position, ice velocities, and discharge of the ice stream and/or outlet glacier (Baldacchino et al., 2022; Horgan et al., 2021; Anandakrishnan et al., 2007). We suggest that changes in basal melting at the KIS grounding zone could modify the driving stresses and velocities at Site 4 by altering the basal friction. Additionally, changes in basal friction reduce the resistive stresses at ice rises downstream of the KIS grounding zone which drive changes

in the velocities at Site 4 and elsewhere on the ice shelf.

### 4.2 Basal melt rates on the Ross Ice Shelf

We can replicate similar intra-annual velocity variations to the GNSS measurements at Sites 1 and 2 by seasonally perturbing basal melt rates with a magnitude of 20 m/a. Our AD-inferred sensitivity map shows that we do not need 20 m/a of perturbation



under the entire ice shelf, but only over 2% of the ice shelf (i.e., identified sensitive regions) to reproduce the intra-annual
velocity variations at Sites 1 and 2. However, is this basal melt perturbation realistic at the identified sensitive regions on the
RIS? In this section, we suggest that the basal melt perturbations we have used in our modelling experiments are realistic for
Sites 1 and 2 but less realistic for Sites 3 and 4.

Firstly, we perturbed the basal melt rates to peak in April and October to reproduce the observed intra-annual velocity
variability on the ice shelf (Figures 3 and 5). Stewart et al. (2019) shows that the timing of these basal melt peaks is realistic for
the RIS near the calving front. Stewart et al. (2019) observes a large basal melt peak in the austral summer (January - March)
> 3 m/a and smaller basal melt peaks in early winter (April and/or May) and late winter (October and/or November) of 1 - 2
m/a at the calving front near the Ross Island pinning point. Additionally, Stewart et al. (2019) observe that the basal melt rates
during winter are "still an order of magnitude higher than the satellite-inferred shelf-wide average and contribute substantially
to the high average melt rates." He suggests that these higher basal melt rates during the early winter are due to the remnant
heat from the summer AASW inflow and in late winter are due to the inflows of HSSW into the ice shelf cavity when large
heat loss and sea ice production leads to active cross-frontal flow that ventilates the cavity (Stewart et al., 2019; Jendersie et al.,
2018; Årthun et al., 2013). These findings support the timing of our perturbed basal melt rate peaks in April and October for
the identified sensitive regions at the calving front near the Ross Island shear zone which influences the velocities at Sites 1, 2,
and 3.

We perturb the basal melt rates near the Ross Island shear zone by a magnitude of 20 m/a for Sites 1 and 2 to reproduce
similar intra-annual velocity variations to the GNSS measurements (Figures 2, 3 and 5). The RIS has low annual average basal
melt rates across the ice shelf (0 - 1 m/a), with the highest average basal melt rates observed at the ice shelf front (> 3 m/a), near
the Ross Island pinning point (Stewart et al., 2019; Stevens et al., 2020; Das et al., 2020; Adusumilli et al., 2020; Schodlok
et al., 2016; Assmann, 2003; Holland et al., 2003; Stern et al., 2013). Figure 6 shows that MITgcm maximum short-term
summer basal melt rates along the calving front are 10 - 25 m/a, which is similar to our basal melt perturbation of 20 m/a for
Sites 1 and 2. Stewart et al. (2019) also observed short-term summer basal melt rates of 10 - 50 m/a at the calving front near
the Coulman High region and Ross Island shear zone. However, these modelled (Figure 6) and observed (Stewart et al., 2019)
basal melt magnitudes of 20 m/a occur during the austral summer, and we perturb basal melt rates in early and late austral
winter.

Summer elevated basal melt rates at the calving front are expected to increase due to continued surface warming and changes
in sea ice cover (Das et al., 2020; Stevens et al., 2020; Stern et al., 2013; Stewart et al., 2019; Horgan et al., 2011; Porter et al.,
2019; Tinto et al., 2019; Moholdt et al., 2014). The basal melt rates have not been observed to increase by the same magnitude
as we have perturbed in the model experiments (20 m/a) in early and late winter, but this may be possible in the future if basal
melt rates continue to increase (Stewart et al., 2019; Porter et al., 2019) and/or changes in primary modes of basal melting occur
(Tinto et al., 2019). Clem et al. (2018) suggests that the switching of the Interdecadal Pacific Oscillation to its positive phase in
the coming decade could increase intrusions of CDW onto the continental shelf. These increased intrusions of the CDW onto
the continental shelf will form mCDW which would reduce sea ice formation in the Ross Sea and enhance basal melting of the
RIS (Tinto et al., 2019; Reddy et al., 2007). Additionally, Greene et al. (2022) suggest that the RIS will experience calving by




the end of the century which may reshape the ocean environment at the ice shelf front and elevate basal melt rates under the
ice shelf.

For Site 3, none of the basal melt perturbations were able to replicate similar intra-annual velocity variations to the GNSS measurements. This suggests that larger basal melt perturbations may be needed at the identified sensitive regions to replicate the observed velocity variability at Site 3. However, basal melt magnitudes of > 80 m/a are highly unrealistic for the RIS today (Adusumilli et al., 2020) and in the future. Therefore, we suggest that Site 3's current observed intra-annual velocity variations
may be driven by a combination of mechanisms not captured by our model.

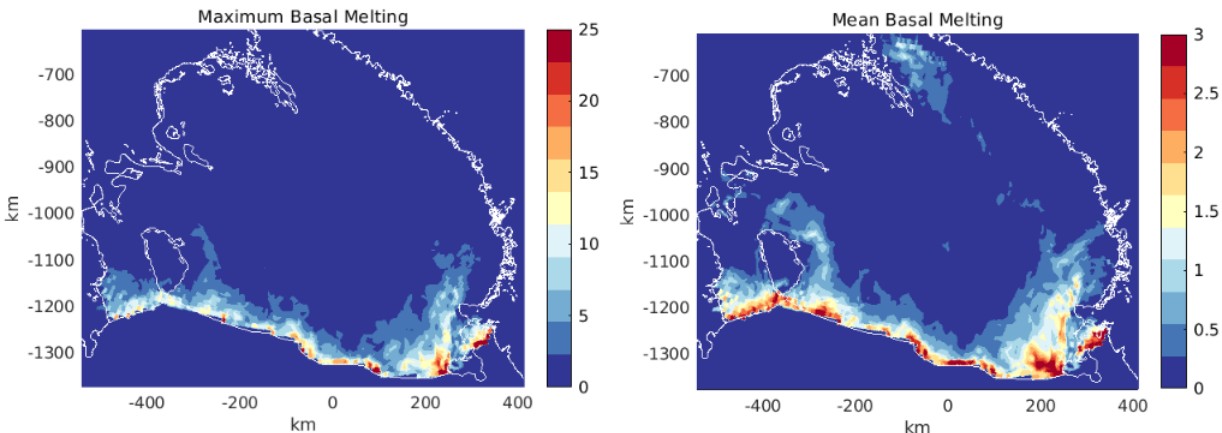

**Figure 6.** MITgcm short-term summer maximum and mean basal melt rates (in m/a) on the RIS.

Additionally, for Site 4 we perturb basal melt rates with magnitudes of 80 m/a at the KIS grounding zone and Siple Coast ice rises to simulate the intra-annual velocity variability. Perturbed basal melt magnitudes of 80 m/a are unrealistically high for the interior of the ice shelf, with observed basal melt rates being low (0 - 1 m/a (Adusumilli et al., 2020)). Localised high basal melt rates of 22.2±0.2m/a have been observed near grounding lines of the Siple Coast Ice Streams (Marsh et al., 2016). However,
no basal melt rates of 80 m/a have been observed on seasonal timescales at the Siple Coast grounding lines or pinning points and ice rises. Seasonally pronounced basal melt predominantly occurs along the RIS front (Jacobs et al., 1992), with Stewart et al. (2019) and Horgan et al. (2011) observing that the seasonal basal melt rates decrease with distance from the calving front. The interior of the ice shelf has a residence time of 1 - 6 years for sub-ice-shelf waters resulting in basal melt rates varying on much longer timescales (Stevens et al., 2020; Reddy et al., 2010; Michel et al., 1979; Smethie Jr and Jacobs, 2005). Therefore,
it is unlikely that Site 4's observed intra-annual velocities are driven by changes in basal melt rates.

We also perturb basal melt rates at the Siple Coast pinning points and ice rises to simulate intra-annual velocity variations at Site 2. It is unlikely that seasonally varying basal melt rates at the Siple Coast pinning points and ice rises are driving the velocity variations at Site 2. We suggest that the seasonal basal melt variability observed at the calving front is driving the majority of Site 2's intra-annual velocity variability. We performed additional experiments where we only perturbed the basal



melt rates at the identified sensitive regions along the calving front close to the Ross Island pinning point. These experiments showed that the phasing and magnitude of velocity variations for Sites 1 and 2 were similar to the experiments presented in this paper, supporting our suggestion that the majority of Site 2's intra-annual velocity variability is driven by seasonal changes in basal melting at the calving front.

### 4.3 Other potential drivers of intra-annual velocity variability

We suggest that seasonal changes in basal melt can explain the observed intra-annual velocity variations at Sites 1 and 2, using perturbed basal melt rates with magnitudes of 20 m/a at the calving front. We also suggest that seasonal changes in basal melt cannot explain the observed intra-annual velocity variations at Sites 3 and 4. We were unable to model similar intra-annual velocity variations to the GNSS measurements for Site 3 using the basal melt perturbations. Additionally, for Site 4, we were able to model similar intra-annual velocity variations using an unrealistically high perturbed basal melt rate of 80 m/a at the KIS

grounding zone and ice rises downstream of the Siple Coast ice streams, where no evidence of seasonally high basal melting rates has been observed. In this section, we discuss other potential drivers of the observed intra-annual velocity variability at Sites 3 and 4.

Most recently, Mosbeux et al. (2023) has shown that the seasonal variability of Sea Surface Height (SSH) modifies ice velocity by changing (1) the driving stress by locally tilting the ice shelf and (2) the basal condition in the grounding zone.

They suggested that SSH variations can lead to changes in bed stresses at the grounding zone, with a negative SSH at the grounding line causing a slow down of ice flow through an increase in basal drag (Mosbeux et al., 2023). Therefore, the intra-annual velocity variations observed at Sites 3 and 4 in this paper could potentially be explained by the seasonal cycle of SSH. However, Mosbeux et al. (2023) did not observe a seasonal velocity variation close to the grounding line of the Whillans Ice Stream when forced by SSH variations, and thus this needs further exploration for the KIS grounding zone.

Additionally, Greene et al. (2018) found that changes in buttressing from sea ice can explain the seasonal cycle of Totten Glacier's ice shelf velocities. Sea ice cover in the Ross Sea decreases in the summer months and increases in the winter months, suggesting that ice shelf velocities would increase in the austral spring and decrease in the austral winter if forced by variations in sea ice backstress (Greene et al., 2018; Cassotto et al., 2015; Howat et al., 2010). However, we observe an acceleration in ice shelf velocities in austral summer and austral winter, indicating that the GNSS velocity variations are not forced by variations

in sea ice backstress.

Seasonal variations in surface air temperatures can also influence the surface melt rates of the ice shelf (Nicolas et al., 2017a; Trusel et al., 2015; Zou et al., 2021a, b) and drive variations in velocities. For example, it has been shown that surface meltwater influences ice shelf velocity by percolating through and weakening the ice shelf shear margins (Cavanagh et al., 2017; Liu and Miller, 1979; Vaughan and Doake, 1996; Greene et al., 2018; Liu and Miller, 1979; Alley et al., 2018). However, the surface

melt rates on the RIS are small, and the response of the ice shelf velocities to summer elevated surface melting has been shown to occur over short timescales (hours to weeks) (Stevens et al., 2022; Chaput et al., 2018; Nicolas et al., 2017a). An El-Niño event occurred in the summer of 2015/2016 when the GNSS measurements for Site 3 were recorded. This event may have increased surface melt rates on the RIS as well as modified wind patterns and ocean circulation (Klein et al., 2020; Paolo et al.,



2015). Nicolas et al. (2017b) observed 14 days of enhanced surface melting on the RIS, between the 10th and 21st of January 2016 due to persistent air temperatures higher than -2°C in the region of Site 3 (Klein et al., 2020; Chaput et al., 2018). These high surface melt rates may have influenced the observed peak in velocities at Site 3 in March 2016 (Figure 3).

Some Siple Coast ice stream velocities are modulated by tidal forcing of the floating ice shelves (Bindschadler et al., 2003; Anandakrishnan et al., 2003; Gudmundsson, 2006). Mosbeux et al. (2023) observed a 6-month tidal signal in their GNSS datasets on the RIS. This 6-month tidal signal may explain the observed intra-annual velocity variability at Site 4. However, Mosbeux et al. (2023) suggests this signal remains a source of noise in their interpretation of intra-annual ice shelf flow rather than a potential driver of ice flow variability. Additionally, significant ice velocity variations due to the diurnal tides of the Ross Sea occur on shorter timescales and closer to the calving front (Brunt et al., 2010; Makinson et al., 2011; Marsh et al., 2013).

Flow variability in the Siple Coast Ice Streams has also been shown to occur on short timescales due to changes in the distribution and supply of basal meltwater (Catania et al., 2012a). Recently, high basal melt rates of 35 m/a have been inferred at the KIS grounding zone within a narrow subglacially sourced basal channel (Whiteford et al., 2022). These high basal melt rates within a subglacial channel suggest that meltwater plumes could be driving changes in the subglacial hydrology system of the KIS. These changes in the subglacial hydrology may be driving variations in the velocities on intra-annual timescales by modifying the basal friction at the KIS grounding line. However, further work is needed to investigate these observed intra-annual velocity variations at Site 4, which is outside the scope of this paper.

## 5 Conclusions

We tested the hypothesis that seasonal variations in basal melt can explain intra-annual velocity variability recorded at four long-duration GNSS stations on the Ross Ice Shelf (RIS) using ISSM. We found that by perturbing basal melt rates at identified sensitive regions on the ice shelf we were able to reproduce similar velocity variations to the GNSS intra-annual velocity variability at two of the GNSS stations. We suggest that seasonal changes in basal melt can explain the observed intra-annual velocity variability at Sites 1 and 2. Our perturbed basal melt rates of 20 m/a on seasonal timescales at the calving front near the Ross Island pinning point are a likely scenario presently and in the future due to continued global warming (Stewart et al., 2019; Stevens et al., 2020; Das et al., 2020). However, we suggest that seasonal changes in basal melting are unable to explain the observed intra-annual velocity variability at Sites 3 and 4. Site 3's modelled velocity variability displays significantly different phasing and magnitudes to the GNSS measurements. We find that the perturbed basal melt rates are unable to reproduce similar velocity variations to the GNSS measurements at Site 3. We suggest that a combination of external forcings may be at play to produce the observed intra-annual velocity variability at Site 3, with recent work by Mosbeux et al. (2023) identifying that changes in SSH can explain this variability. Finally, we also suggest that seasonal changes in basal melt are unable to explain the observed intra-annual velocity variation at Site 4, with the unrealistic basal melt perturbation of 80 m/a needed to produce similar magnitudes in velocity variations. There have been no observations of seasonal changes in basal melt rates at the Siple Coast ice streams grounding lines, with the interior of the ice shelf experiencing much lower basal melt rates compared to the calving front (Horgan et al., 2011; Adusumilli et al., 2020; Smethie Jr and Jacobs, 2005). We suggest that Site 4's observed



intra-annual velocity variability is driven by a combination of external forcings and internal mechanics modifying the driving stresses and basal friction at the grounding line.

Future work should focus on (1) continuing the multi-year GNSS records of seasonally resolved ice velocity changes on the RIS, (2) providing further understanding of the ice shelf interactions with basal melt rates on floating and grounded ice through coupled ocean-ice shelf models, and (3) exploration of other potential drivers of intra-annual velocity variations particularly concerning the observed seasonal velocity variations at the KIS grounding zone.

Our results can provide a further understanding of potential mechanisms driving the intra-annual velocity variations on the RIS. The observed intra-annual velocity variations at all GNSS sites are likely driven by a complex combination of external forcings and internal mechanics on the RIS. However, our sensitivity maps highlight where increases in basal melt rates will influence ice velocity today and in the future. Specifically, the sensitive regions identified at the calving front near the Ross Island pinning point are already undergoing significant seasonal increases in basal melt rates (Stewart et al., 2019). Our AD-inferred sensitivity map shows that we do not need 20 m/a of perturbation under the entire ice shelf, but only over 2% of the ice shelf to reproduce the intra-annual velocity variations. Therefore, we are likely to observe continued intra-annual velocity variations on the RIS driven by seasonal changes in basal melting at the calving front.





## Appendix A

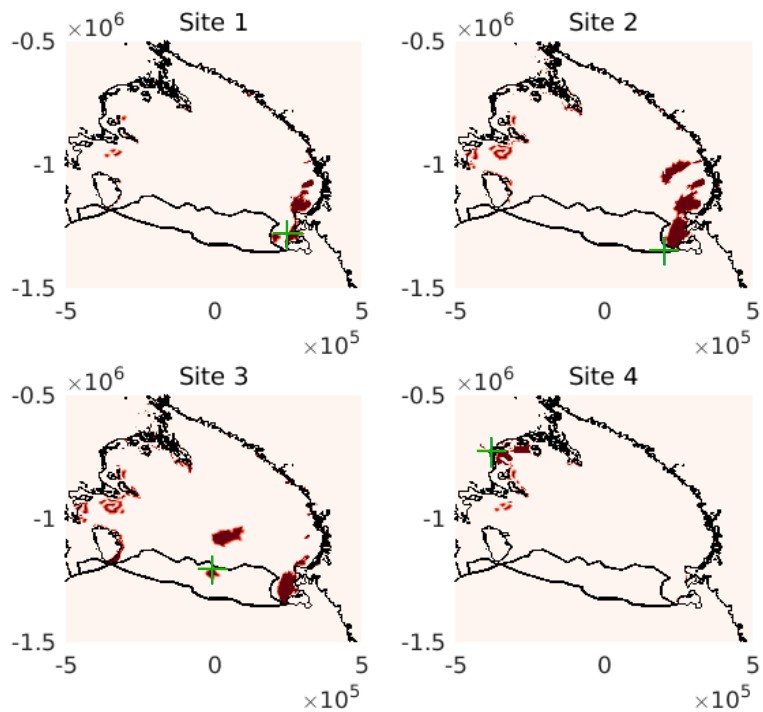

**Figure A1.** Locations where the basal melt rate is perturbed seasonally (i.e., where the AD mapped sensitivity is greater than $0.5e-11$ m$^{-2}$ (highlighted in dark red)) for each GNSS site (green marker). The passive ice on the RIS identified by Fürst et al. (2016) is also outlined in black.

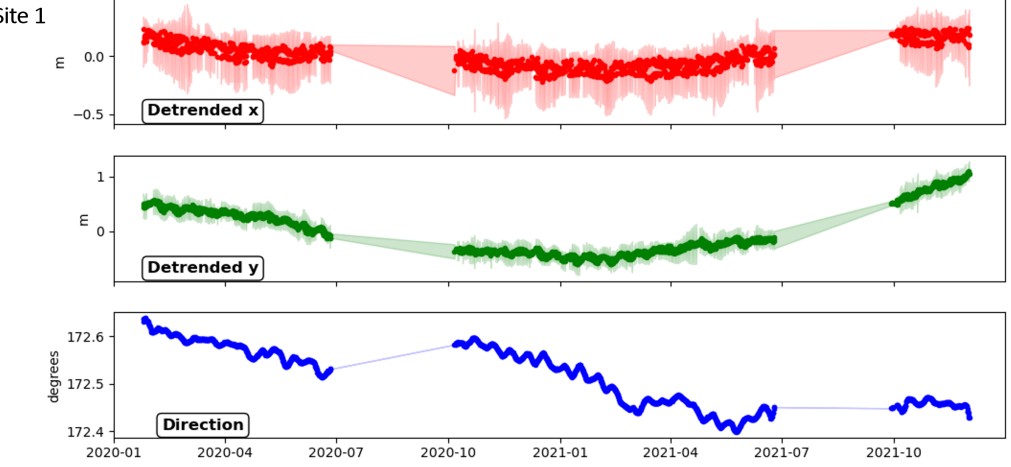

**Figure A2.** Site 1 GNSS detrended position (x, y) and direction (clockwise from from grid north).





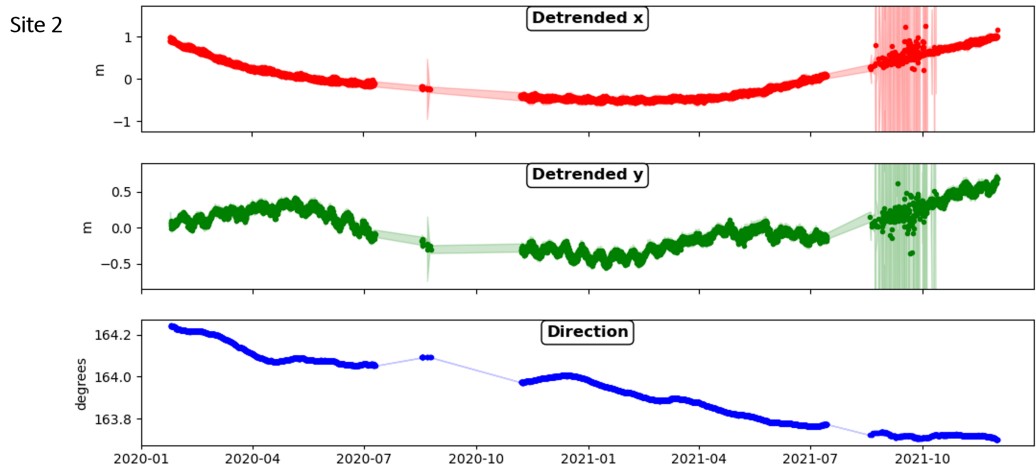

**Figure A3.** Site 2 GNSS detrended position (x, y) and direction (clockwise from from grid north).

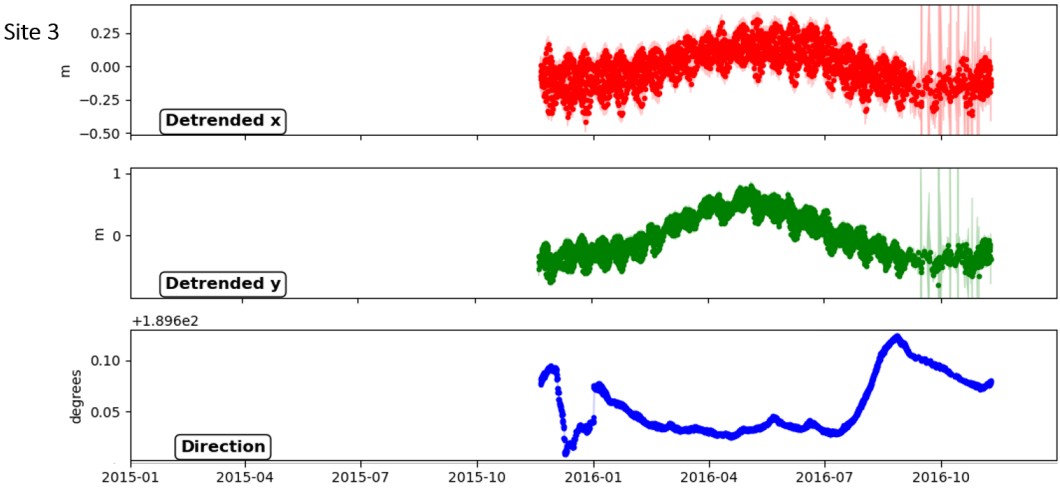

**Figure A4.** Site 3 GNSS detrended position (x, y) and direction (clockwise from from grid north).



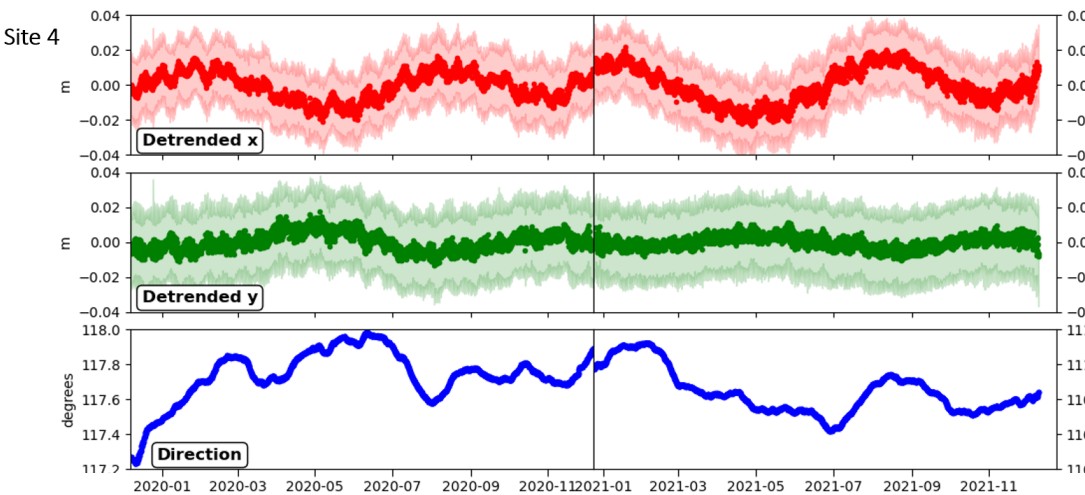

**Figure A5.** Site 4 GNSS detrended position (x, y) and direction (clockwise from from grid north).



*Data availability.* The Ice-sheet and Sea-level System Model v4.18 can be accessed at https://issm.jpl.nasa.gov. BedMachine Antarctica is available at NSIDC (http//nsidc.org/data/nsidc-0756). InSAR-Based ice velocity is found at NSIDC (https://nsidc.org/data/nsidc-0484/). The Antarctic surface mass balance (RACMO 2.3p2) is available at https://www.projects.science.uu.nl/iceclimate/models/racmo-data.php. GNSS
data will be uploaded to a suitable repository (Zenodo) on acceptance.

*Author contributions.* Project vision, funding acquisition and field planning (2019/20 season)was led by Nicholas Golledge. The event leader for field season 2021/2022 was Alanna V. Alevropoulos-Borrill. Field assistants for fieldwork included Francesca Baldacchino, Laurine van Haastrecht, Dan Lowry, and Alexandra Gossart. Site 4 data acquisition and GNSS processing of all sites was carried out by Huw Horgan. GNSS analysis was carried out by Francesca Baldacchino and Huw Horgan. ISSM simulations and AD analysis were carried out by Francesca
Baldacchino under the guidance of Mathieu Morlighem. AD simulations were carried out by Mathieu Morlighem. MITgcm basal melt outputs were provided by Alena Malyarenko. All authors contributed to the manuscript.

*Competing interests.* Huw Horgan is an associate editor for The Cryosphere and EGUsphere. The authors declare that they have no other conflicts of interest.

*Acknowledgements.* We are grateful to Antarctic New Zealand for facilitating and supporting our field campaigns to the Ross Ice Shelf to
install and collect GNSS measurements. Additionally, we are grateful to Peter Bromirski for sharing the raw GNSS datasets collected on the Ross Ice Shelf in 2015-2016 that have been included in this paper. Francesca Baldacchino thanks Laurie Padman for helpful correspondence early in this study. This work was supported by the New Zealand Ministry for Business, Innovation and Employment contracts RTUV1705 ("NZSeaRise") and ANTA1801 ("Antarctic Science Platform"), and Royal Society of New Zealand contracts VUW-1501 and VUW16-02.



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
