# Peer review of "Modelling GNSS-observed seasonal velocity changes of the Ross Ice Shelf, Antarctica, using the Ice-sheet and Sea-level System Model (ISSM)"

_EGUsphere, 2023_

## Referee Comment (RC3)

**Review of "Modelling GNSS-observed seasonal velocity changes of the Ross Ice Shelf, Antarctica, using the Ice-sheet and Sea-level System Model" by Baldacchino et al.**

Over the last decade, several GNSS stations have been deployed across the Ross Ice Shelf, revealing strong signals with noticeable variability on intra-annual time scales, suggesting a seasonal pattern. This variability in horizontal displacement can be expressed as variations in the ice flow velocity.

Klein et al. (2020) suggested that seasonal changes in basal melt might contribute to this variability, but their modeled velocities based on realistic seasonal melt rates were much smaller than observed values (i.e. about 10% of the observed variations). Similarly, Mosbeux et al. (2023) explored how seasonal fluctuations in sea surface height could affect ice velocity through changes in driving stress over the ice shelf and changes in basal stresses at the grounding zone.

In this manuscript, the authors also investigated the effect of the seasonality in basal melt on the seasonality in ice flow velocities using ISSM. They first use Automatic Differentiation (AD) tools built in the ISSM model to pinpoint regions where changes in basal melt rate have the most influence on the ice flow.

Regions highly sensitive to seasonal melting align closely with those identified in studies focusing on the effect of buttressing on the grounded ice dynamics (Furst et al., 2016; Reese et al., 2018). Notably, the vicinity of Ross Island, known for significant buttressing effects, exhibits high sensitivity to basal melt. This same region has also been shown to exhibit large summer melt rates, as supported by both modeling (Tinto et al., 2019) and observational studies (Steward et al., 2019).

Similarly, to Klein et al. (2020), the authors then use an ocean model to explore the sensitivity of the flow to a realistic melt rate pattern. Although the authors utilize MITgcm melt rates instead of the ROMS melt rates from Klein et al. (2020), both highlight a similar melt rate pattern. However, the present study employs significantly higher melt rates by amplifying their model melt rates following a simple sinusoidal function over time in high sensitivity regions highlighted by the AD. With their method, they find that velocity variations match the observations only when basal melt rates peak at over 80 m/a on the top of the background signal of MITgcm. Such melt rates appear disproportionately high compared to the observed 3 m/a reported by Steward et al. (2019) or their own MITgcm outputs. While Klein et al. demonstrated that by increasing their melt rates by a factor of ten (reaching roughly 20 m/a) they could better match observed velocity changes at GNSS stations, they could not justify such melt rates based on current observations and ocean modelling. There is also no evident that he amplified melt rates used by the authors show realistic patterns both in time and space.

I therefore have several concerns regarding the realism of the modeled melt rates and the conclusions of the paper. Furthermore, the paper overlooks the potential influence of other factors such as sea surface height variations and tidal effects, which have been shown to significantly impact ice flow dynamics in previous research. Even focusing solely on basal melt rates, seasonal melt close to the grounding line where ocean models usually struggle to

correctly model high melt rates (e.g., the melt under Pine Island ice shelf in Dutrieux et al., 2013) and their effect on the grounding zone, could have been explored by the authors.

**Specific comments**

- Figure 1: To me, this figure could be reworked and made cleaner. Why drawing null velocities in the ocean? It only decreases the readability.

- On site 3, which is the main site used by Klein et al. (2020), the data derivation from displacement to velocities gives you a minimum in April.

- Figure 2. The figure really looks like a draft and not a publishable figure. The grounding line and the safety bands are both plotted in black. There is no metrics on the x and y that are used and written. The southern part of the grounding line is cutoff without specific reasons.

- Section 2.4: You propose a time varying melt rate $M_b(t)$ based on MITgcm output with an additional perturbation based on a sinusoidal variation $\sin 4\pi + 3t$ in locations where the ice flow shows a large sensitivity to melt rates variations. The amplitude of the sinusoidal perturbation is defined by a factor $p$. Later, we quickly realize that MITgcm variations do not affect the flow at all (variations much lower than 1 m/a in all sites, as shown in Figure 5). This means that the seasonality of MITgcm does not affect significantly the ice flow. The only way to trigger significant variations is to apply a sufficient perturbation $p \sin 4\pi + 3t$. However, this perturbation is not based on any realistic ocean simulation or observations and even seems to go against the model, as stated by the authors *"The raw MITgcm basal melt rates displayed a seasonal signal, however, the amplitude of this seasonal variability was not large enough and the phasing incorrect to replicate the GNSS observed velocity variability."*

  To me, if the MITgcm modelling shows a seasonality in melt rates, this seasonality should be explored, even if it does not give the correct phasing on the ice flow velocities. The MITgcm melt rates should be shown with maps of melt rates at different period of the years, or at least with a timeseries of the integrated melt rates over the ice shelf. For example, the model melt rates in Klein et al. (2020) shows only one peak melt rate in February (see their Figure 7a or the maps in Fig. 8). Why building a twice peaking melt rate if it is not realistic or backed by any modelling or observation?

- Figure 5: Looking at the pattern of your observed velocity variations, it seems that ice flow reaches a minimum velocity in March and a second one in August. My understanding is that this is the reason why the authors apply two peak melt rates in your idealized sinusoidal melt. However, such a semi-annual cycle caused by something different like a semi-annual-variability in tidal amplitudes and affecting the grounding zone of the ice shelf, as suggested in Mosbeux et al. (2023) conclusions. This could be seen as a process similar to the non-linear response of the ice shelf (and the ice sheet) to the diurnal tide (e.g. Gudmnundsson, 2011; Rosier et al., 2020).

  Site 3 semi-annual cycle does not seem as clean as on other sites but still visible with a sharp drop in velocity in November followed by plateau from early January to March, a second drop in March-April before a reversal with a speed up until August, ending with a second Plateau from August to November. From the detrended displacement in Figure A4,

we do not see any sharp change in displacement in November. How do you explain such result? Also, the strong direction changes before January 2016, does not really reflect in the detrended x and y displacement. Looking at Klein et al. (2020), the velocity trend looks a bit different. It would be good to investigate the reasons for this.

**Bibliography**

Dutrieux, P., Vaughan, D. G., Corr, H. F. J., Jenkins, A., Holland, P. R., Joughin, I., and Fleming, A. H.: Pine Island glacier ice shelf melt distributed at kilometre scales, The Cryosphere, 7, 1543–1555, https://doi.org/10.5194/tc-7-1543-2013 , 2013.

Furst, J. J., Durand, G., Gillet-Chaulet, F., Tavard, L., Rankl, M., Braun, M., and Gagliardini, O.: The safety band of Antarctic ice shelves, Nature Climate Change, 6, 479–482, https://doi.org/10.1038/nclimate2912, 2016.

Gudmundsson, G. H.: Ice-stream response to ocean tides and the form of the basal sliding law, The Cryosphere, 5, 259–270, https://doi.org/Gudmundsson, G.H.. 2011 Ice-stream response to ocean tides and the form of the basal sliding law. The Cryosphere, 5 (1). 259-270. https://doi.org/10.5194/tc-5-259-2011 <https://doi.org/10.5194/tc-5-259-2011>, 2011.

Reese, R., Gudmundsson, G. H., Levermann, A., and Winkelmann, R.: The far reach of ice-shelf thinning in Antarctica, Nature Climate Change, 8, 53–57, https://doi.org/10.1038/s41558-017-0020-x, 2018.

Rosier, S. H. R. and Gudmundsson, G. H.: Exploring mechanisms responsible for tidal modulation in flow of the Filchner–Ronne Ice Shelf, The Cryosphere, 14, 17–37, https://doi.org/10.5194/tc-14-17-2020, 2020.

---

## Author Comment (AC1)

**Modelling GNSS-observed seasonal velocity changes of the Ross Ice Shelf, Antarctica, using the Ice-sheet and Sea-level System Model (ISSM)**
**– Response to Reviewer 1 –**

Francesca BALDACCHINO et al

March 15, 2024

Firstly, we would like to thank all three reviewers for their constructive and detailed comments. We agree with many points that were raised, especially the lack of discussion of our results in the context of recently published papers (e.g., *Klein et al.* [2020] & *Mosbeux et al.* [2023]). We have responded to each reviewer's comments below. All three reviewers' main comments included the need for additional discussion and consideration of *Klein et al.* [2020] and *Mosbeux et al.* [2023]. This is a good point that we will address throughout a revised manuscript, as we realize that we did not adequately motivate and contextualise our study, and these previous works deserved more recognition.

**Novel contributions**

Several reviewers questioned the novelty of aspects of our study. Here we briefly summarise aspects of our study which we believe are novel contributions.

- We present new GNSS time series from the Ross Ice Shelf that have not previously been published. These include sites near the calving front, near a significant pinning point, and in the deep interior of the ice shelf near the grounding line.

- Notably, our Site 2 is close to the calving front near the Ross Island region, which has been identified as observing high basal melt rates on a seasonal timescale (Stewart et al., 2019).

- We show that these measurements consistently show 2 peaks in ice shelf velocity every year (for Sites 1, 2 and 4, the newly collected datasets), contrary to previous measurements presented in *Klein et al.* [2020] and *Mosbeux et al.* [2023].

- We suggest that the seasonal variability of SSH (i.e., yearly cycle) may not be able to reproduce our GNSS seasonal velocity variability (i.e., semi-annual).

- We therefore turned to the potential role of basal melt and wanted to test *what it would take* to match velocity variations by changing the forcing as little as possible.

- Our approach of combining Automatic Differentiation and weekly MITgcm basal melt rates ([*Klein et al.*, 2020] used monthly basal melt rates) is also novel.

**Sea surface height (SSH)**

An area where the two previous studies should be discussed more in our manuscript is in regards to what other factors that could be driving the observed velocity variations on the ice shelf. [*Mosbeux et al.*, 2023] nicely shows that the seasonal variability of SSH can explain their observed seasonal variability of ice velocity.

To take this into account, we will rerun our simulations with the same SSH forcings implemented in [*Mosbeux et al.*, 2023] to consider this factor. However, we expect that the seasonal variability of SSH cannot explain our two-peaked seasonal velocity variability, as mentioned above. In our revised manuscript, we will also discuss other possible factors (tides, sea ice buttressing etc), that may also be good candidates to explain our new GNSS observations.

**Basal melt rates**

Melt rates are difficult to model and properly constrain, especially close to grounding lines, despite their critical role on ice dynamics. All reviewers commented on the realism of the basal melt rates. We agree that these basal melt rate perturbations we use are extremely high for the Ross Ice Shelf, today and in the future. However, this paper focuses on asking whether perturbations in basal melt rates *can* reproduce a similar velocity variability as observed by the GNSS units. We acknowledge that our contribution is a proof of concept, not a definitive answer to the question, and we will do our best to make this clear in the revision.

**Multiple peaks in melt rate perturbation**

Several reviewers questioned our use of multiple peaks in melt rate perturbation. Here we clarify our motivation for doing so. The baseline weekly MITgcm basal melt rates include a clear peak in the austral summer, and multiple other (much smaller) peaks throughout the year, highlighting that the basal melt rates have more variability than presented in [*Klein et al.*, 2020]. We also refer to [*Stewart et al.*, 2019] basal melt observations in our discussion, highlighting that they observe the largest peak in the austral summer, but also smaller peaks in the austral winter.

*Klein et al.* [2020] suggest that the actual total summer increase in the heat content of the AASW layer near the ice front is likely to be larger than the modelled increase, and the seasonal enhancement of the basal melting will continue further into autumn than in their model. [*Klein et al.*, 2020] extended the late melt period to April and found that it also shifted the timing of maximum velocity a month later, showing that a longer or later melt period at the front could align the modelled and observed velocity phases.

Our approach is to use multiple basal melt peaks as the basis for our phasing of the basal melt forcing, and we apply perturbations on this forcing until we reproduce a similar velocity variability to the GNSS observations. Through this, we can highlight that seasonal basal melt rates can reproduce the GNSS velocity variability on an interannual timescale for XX of the sites. We do not state or intend to imply that these perturbed basal melt rates are realistic for the Ross Ice Shelf. Our study instead serves as a proof of concept, motivated by Klein et al. "as-yet-unidentified seasonal processes". This overall aim will be clarified in the revised manuscript.

**1 Reviewer 1**

**1.1 General comments**

As stated in the submission guidelines for The Cryosphere, I strongly encourage the authors to deposit all scripts and configuration files for setting up and running the ISSM simulations in a FAIR-aligned repository, such as Zenodo.

We will include the scripts and configuration files for setting up and running the ISSM simulations in an FAIR-aligned repository when submitting the final manuscript.

Add to the introduction paragraph on lines 80-88 a brief description of how this present study differs from [*Klein et al.*, 2020] and [*Mosbeux et al.*, 2023] which, as described in the preceding paragraph, provide an explanation for the intra-annual velocity variations for the RIS.

This is a great point, see summary above.

Additionally, please add text in the Discussion that addresses why the conclusions of this present study differ from [*Klein et al.*, 2020], which claim that seasonal velocity variations are not driven by basal melt rate variability.

Again, this is a good point, hopefully addressed in the summary above.

Use "intraannual" throughout the text to refer to monthly to seasonal variability. For example, line 73 refers to monthly to seasonal variability as "interannual" but this should be changed to "intraannual". Please check the entire manuscript for other cases of this.

Done, this has been edited throughout the manuscript.

Add a map of observed velocities of the ice shelf to Figure 1.

Figure 1 shows the modelled surface velocity results after initialisation, we think that displaying the modelled surface velocity results is sufficient since ISSM used observed velocity from the MEaSUREs dataset to calibrate the model (modeled and observed velocity therefore look similar).

Remove Figure 3 because Figure 5 shows the same data.

We think that Figure 3 should still be included as it shows the raw *observed* velocities directly from the GNSS sites, whereas Figure 5 highlights the velocity variations. The observed velocities are important to show as they highlight at which sites the velocities are flowing faster, and the seasonal changes in these velocities.

It is not clear to me whether including both Figure 2 and Figure 4 is necessary and how the interpretation differs for the results shown in these two Figures. My understanding is that Figure 2 shows the sensitivity of the final velocity for a 6-month simulation and Figure 4 shows the sensitivity of the final velocity for a 2-year simulation. I also see that Figure 2 shows sensitivities that are above the selected threshold, whereas Figure 4 shows the full range of sensitivities. However, it seems like the text in Section 3.2, which describes the results in Figure 4, could also apply to the results in Figure 2. I may be wrong, in which case please feel free to disagree. If it is decided to keep both figures in the manuscript, please add text to Section 3.2 that explains why the results in the two figures are different and what additional information for interpretation is provided by Figure 4 that isn't already provided by Figure 2.

We agree with the reviewer that Figure 2 is not necessary, it will be removed.

Add a figure showing absolute modelled and observed velocities at each GNSS site to Supplementary Materials and reference this figure on line 266.

This figure will be added to the Supplementary Materials, although we are focusing on velocity variations, not absolute velocities.

The paragraph on lines 353-360 hypothesizes that perturbing the melt rate at the KIS grounding zone could modify driving stress at Site 4, through a modification of basal friction. Couldn't you use the ice sheet model to test this proposed process? ISSM simulates changes in driving stress and the corresponding change in basal friction due to the simulated melt rate perturbations. You could analyze the changes in the force balance at Site 4 to address this. Please either add this analysis to the paper or provide text explaining why this is not possible with your model configuration.

We will model the driving stress through time at Site 4, and analyse the results. If we think they are appropriate to include in the updated manuscript, we will add a figure and explanation in the discussion.

Wherever possible, begin each paragraph in the Discussion section with a topic sentence that describes the main result that is being discussed in the paragraph. For example, on line 312, change the topic sentence to: "We model a seasonal signal in velocity variability that is similar in phasing and magnitude at GNSS Sites 1 and 2 but not Sites 3 and 4." Another example is on line 362, where the topic sentence could be changed to: "The melt rate perturbations used in our modelling experiments are realistic for Sites 1 and 2 but less realistic for Sites 3 and 4." Please go through the Discussion to find other opportunities to make changes to topic sentences to clarify the result being discussed in the paragraph.

This is a great suggestion, and we will go through the discussion and begin each paragraph with a topic sentence in the updated manuscript.

**1.2 Specific comments**

[line 13] The word "today" seems out of place here. Can it be removed?

Done.

[line 126] I suggest adding a reference to https://doi.org/10.5623/geomat-2005-0004 to cite the CSRS-PPP specifically.

Done.

[line 175] Add a sentence to Section 2.4 stating that one set of simulations was run in which the basal melt rates were perturbed at locations where there was sensitivity in the velocities for any of the GNSS locations (as opposed to separate simulations where the melt rates were perturbed for each individual GNSS location).

We did not do this, and we apologize for the confusion. We only ran *one* set of simulations in which the basal melt rates were perturbed at locations identified as highly sensitive (using our threshold) for at least one GNSS site. We will clarify this in Section 2.4.

[line 181] Replace "raw" with "unperturbed".

Done.

[Figures 2 and 4] Is the black line showing the grounding line? If so, state that in the caption and change the passive ice outline from black to a different colour.

The caption has been edited to state that the black line is the grounding line. These figures will be edited in the updated manuscript, with a different colour for the passive ice outline.

[Figure 4] Add labels and arrows showing the locations of Roosevelt Island, Crary Ice Rise, Steershead Ice Rise, the Shirase Coast Ice Rumplus, Byrd Glacier and any other locations that are refered to in the text when describing this figure.

Thank you for this suggestion, we will add labels to Figure 1 to identify the locations that are referred to in the text in the updated manuscript.

[Figure 5] Color each dotted black line using the same colors as the solid lines to denote the melt rate perturbation magnitudes.

This will be implemented in the updated manuscript.

[line 264] The text states that "use of the lower sensitivity value did not significantly affect the final modelled velocity variations" and Figure 5 shows that is indeed correct for sites 1-3 but for site 4, it looks like the velocity peaks are about 30 percent larger for the highest perturbation magnitude. I suggest quantifying the differences between the two simulations that used different sensitivity thresholds.

While we agree with the reviewer that this statement should be toned down, we are a bit resistant to play with this threshold too much as the aim of this paper is not to try and match the GNSS and modelled velocities, but rather we are trying to determine if it is possible to reproduce similar velocity variations from basal melt alone, at least for some GNSS sites.

[lines 340-341] This sentence needs to be reworded: "Figure 2 highlights that basal melt rates are perturbed at the Ross Island region for Sites 1, 2 and 3." to something like "Figure 2 shows that velocities at Sites 1, 2, and 3 are most sensitive to basal melt rate perturbations at the Ross Island region."

Done.

[lines 350-352] This sentence is repetitive and states the same thing as the previous paragraph. Please delete this.

Done.

[line 361] Change this section heading to: "Comparison to observed basal melt rates beneath the Ross Ice Shelf".

Done.

[line 384] Please define what "short-term" basal melt rates are. This is the first mention of this term and it is not clear how this is defined.

The term "short-term" has been removed.

[Figure 6] Similar to the previous comment, please define the timespans that "short-term" and "mean" are covering in the figure caption.

The term "short-term" has been removed and the caption has been edited to replace "mean" with "average".

[lines 419-423] Are these "additional" experiments ones that are already described in the paper? If not, the configuration and results from these additional experiments need to be included in paper. They can be added to Supplementary Materials or an Appendix but please add figures of (1) the locations where melt rates were perturbed and (2) the resulting velocity variations.

These additional experiments will be included in the Appendix alongside the figures.

**References**

Klein, E., C. Mosbeux, P. D. Bromirski, L. Padman, Y. Bock, S. R. Springer, and H. A. Fricker, Annual cycle in flow of Ross Ice Shelf, Antarctica: Contribution of variable basal melting, *Journal of Glaciology*, *66*(259), 861–875, doi:10.1017/jog.2020.61, 2020.

Mosbeux, C., L. Padman, E. Klein, P. Bromirski, and H. Fricker, Seasonal variability in antarctic ice shelf velocities forced by sea surface height variations, *The Cryosphere*, *17*(7), 2585–2606, 2023.

Stewart, C. L., P. Christoffersen, K. W. Nicholls, M. J. Williams, and J. A. Dowdeswell, Basal melting of Ross Ice Shelf from solar heat absorption in an ice-front polynya, *Nature Geoscience*, *12*(6), 435–440, doi:10.1038/s41561-019-0356-0, 2019.

---

## Author Comment (AC2)

**Modelling GNSS-observed seasonal velocity changes of the Ross Ice Shelf, Antarctica, using the Ice-sheet and Sea-level System Model (ISSM)**

- Response to Reviewer 2-

**Francesca BALDACCHINO et al**

March 15, 2024

Firstly, we would like to thank all three reviewers for their constructive and detailed comments. We agree with many points that were raised, especially the lack of discussion of our results in the context of recently published papers (e.g., *Klein et al.* [2020] & *Mosbeux et al.* [2023]). We have responded to each reviewer's comments below. All three reviewers' main comments included the need for additional discussion and consideration of *Klein et al.* [2020] and *Mosbeux et al.* [2023]. This is a good point that we will address throughout a revised manuscript, as we realize that we did not adequately motivate and contextualise our study, and these previous works deserved more recognition.

**Novel contributions**

Several reviewers questioned the novelty of aspects of our study. Here we briefly summarise aspects of our study which we believe are novel contributions.

- We present new GNSS time series from the Ross Ice Shelf that have not previously been published. These include sites near the calving front, near a significant pinning point, and in the deep interior of the ice shelf near the grounding line.
- Notably, our Site 2 is close to the calving front near the Ross Island region, which has been identified as observing high basal melt rates on a seasonal timescale (Stewart et al., 2019).
- We show that these measurements consistently show 2 peaks in ice shelf velocity every year (for Sites 1, 2 and 4, the newly collected datasets), contrary to previous measurements presented in *Klein et al.* [2020] and *Mosbeux et al.* [2023].
- We suggest that the seasonal variability of SSH (i.e., yearly cycle) may not be able to reproduce our GNSS seasonal velocity variability (i.e., semi-annual).
- We therefore turned to the potential role of basal melt and wanted to test *what it would take* to match velocity variations by changing the forcing as little as possible.
- Our approach of combining Automatic Differentiation and weekly MITgcm basal melt rates ([*Klein et al.*, 2020] used monthly basal melt rates) is also novel.

**Sea surface height (SSH)**

An area where the two previous studies should be discussed more in our manuscript is in regards to what other factors that could be driving the observed velocity variations on the ice shelf. [Mosbeux et al., 2023] nicely shows that the seasonal variability of SSH can explain their observed seasonal variability of ice velocity.

To take this into account, we will rerun our simulations with the same SSH forcings implemented in [Mosbeux et al., 2023] to consider this factor. However, we expect that the seasonal variability of SSH cannot explain our two-peaked seasonal velocity variability, as mentioned above. In our revised manuscript, we will also discuss other possible factors (tides, sea ice buttressing etc), that may also be good candidates to explain our new GNSS observations.

**Basal melt rates**

Melt rates are difficult to model and properly constrain, especially close to grounding lines, despite their critical role on ice dynamics. All reviewers commented on the realism of the basal melt rates. We agree that these basal melt rate perturbations we use are extremely high for the Ross Ice Shelf, today and in the future. However, this paper focuses on asking whether perturbations in basal melt rates *can* reproduce a similar velocity variability as observed by the GNSS units. We acknowledge that our contribution is a proof of concept, not a definitive answer to the question, and we will do our best to make this clear in the revision.

**Multiple peaks in melt rate perturbation**

Several reviewers questioned our use of multiple peaks in melt rate perturbation. Here we clarify our motivation for doing so. The baseline weekly MITgcm basal melt rates include a clear peak in the austral summer, and multiple other (much smaller) peaks throughout the year, highlighting that the basal melt rates have more variability than presented in [*Klein et al.*, 2020]. We also refer to [*Stewart et al.*, 2019] basal melt observations in our discussion, highlighting that they observe the largest peak in the austral summer, but also smaller peaks in the austral winter.

*Klein et al.* [2020] suggest that the actual total summer increase in the heat content of the AASW layer near the ice front is likely to be larger than the modelled increase, and the seasonal enhancement of the basal melting will continue further into autumn than in their model. [*Klein et al.*, 2020] extended the late melt period to April and found that it also shifted the timing of maximum velocity a month later, showing that a longer or later melt period at the front could align the modelled and observed velocity phases.

Our approach is to use multiple basal melt peaks as the basis for our phasing of the basal melt forcing, and we apply perturbations on this forcing until we reproduce a similar velocity variability to the GNSS observations. Through this, we can highlight that seasonal basal melt rates can reproduce the GNSS velocity variability on an interannual timescale for XX of the sites. We do not state or intend to imply that these perturbed basal melt rates are realistic for the Ross Ice Shelf. Our study instead serves as a proof of concept, motivated by Klein et al. "as-yet-unidentified seasonal processes". This overall aim will be clarified in the revised manuscript.

**1 Reviewer 2**

**1.1** General comments**

The findings of Klein et al., 2020 and Mosbeux et al., 2023 are referenced but then appear to be largely disregarded until the discussion. In particular, the potential influence of changes in sea surface height and ocean tides is completely overlooked. Tides are known to cause substantial variations in velocity over short periods (e.g. Anandakrishnan et al., 2003; Doake et al., 2002), and also potentially over long periods of up to a year (e.g. Murray et al., 2007). The GNSS processing smooths out short-term tidal effects, but I expect daily variability is large, being previously observed nearby at up to 100 percent of the mean (e.g. Brunt et al., 2010). It is not inconceivable that small, solar annual or semi-annual tides could drive the remaining j1 percent semi-annual variations in velocity shown in Figure 3 and it needs to be explained why they can be ignored.

Hopefully our summary above addresses this important and fair comment. In the revision, we will re-run our model experiments with the seasonal variability in SSHs.

Secondly, it is not clear to me why a more realistic melt forcing was not used? The forcing used here (and required to match behaviour at sites 1 and 2) is symmetric with two peaks, when observations from Stewart et

al., 2019 show only one dominant peak in the northwest region near Ross Island. Also, the peak melting observed by Stewart et al., 2019 occurs consistently in February while here melt peaks are applied in April and October, and the magnitude of the melt perturbations appears to be substantially higher than observations, with 20 m/a additional melt required, presumably on top of a baseline rate?

Again, this is a fair point. We realize that we have not made the objective of our study clear. Our experiments try to quantify how high the melt rates need to be to explain the variations in ice speed. We highlight that these perturbed basal melt rates are unrealistic for Sites 3 and 4, and suggest that they may be more realistic for Sites 1 and 2 and are potentially likely to occur in the future due to global warming. Stewart et al., 2019 found maximum summer basal melt rates of 52 m/a at the calving front near Site 2, highlighting that our basal melt magnitudes are not that unrealistic compared to the current maximum summer basal melt rates. Please refer to the summary provided above for a more detailed response to this comment. We will also clarify that these perturbed basal melt rates are on top of the baseline rate provided by MITgcm, and we will provide a figure of these baseline basal melt rates in the Appendix.

My fundamental issue with the paper is that without an understanding or attempt to account for other more dominant factors that are driving variability, I don't think it can be concluded with any confidence that seasonal melt is driving (or even influencing) seasonal velocity variability at any of the sites.

Hopefully, our responses above clarify this point, we realise now that we did not take into account other studies in our current manuscript, and this will be updated significantly.

Additionally, other possible dominant factors will be discussed in more detail in the updated manuscript and we will make sure to account for changes in SSH, following *Mosbeux et al.* [2023].

**1.2** Specific comments**

Abstract: 'sensitive regions' are mentioned several times before it can be determined that this means regions where ice velocities are sensitive to basal melt

The sensitive regions being discussed here are related to *Baldacchino et al.* [2022]; *Fürst et al.* [2016]; *Gudmundsson et al.* [2019]; *Reese et al.* [2018]. We will clarify this in the updated manuscript.

Line 12: "We suggest that... velocity variations... could be partly.... driven by melt". This is a very tentative conclusion and implies that you don't believe this to be the case either.

This is a sentence where we tried to clarify the scope of the study, which is to investigate whether basal melt CAN explain GNSS' data, but we are not providing a definite answer.

Line 55: sp. Siple Coast ice streams

Done.

Line 70: I would argue that GNSS doesn't provide a unique opportunity to measure seasonal variations in velocity – satellite methods can also do this – although GNSS does have better resolution and accuracy.

We would argue that the high temporal resolution of the GNSS instruments that are deployed for multiple years on the ice shelf still provide this unique opportunity. Furthermore, GNSS allows us to capture wintertime velocities, when visible-band remote sensing is not possible (due to 24hr darkness in Antarctica). This alone makes the data unique.

Line 70: sp. MacAyeal

Done.

Line 80: While you discuss the work of Klein et al., and Mosbeux et al. in the previous paragraph, the aims and approach taken in this paper do not follow from this discussion.

This section will be edited so that the aims and approach in this manuscript follow on from the discussion.

Line 101-104: These two sentences contradict each other.

This sentence has been removed "However, thinning of this region or changes in the ice-front location (i.e., via iceberg calving) will alter the stress balance and velocities of the ice shelf (Gudmundsson et al., 2019; Klein et al., 2020)."

Line 129: What is meant by 'data processing was iterated'? How are initial positions updated and until what criteria are met?

Because initial position precision wraps into final position precision, updating the initial position in a GNSS Rinex file with an estimate closer to the actual initial position improves the final estimate. To achieve this, the data is processed the first time, this first estimate of position is then used as the initial position for a second processing round. This is accomplished by creating a new rinex GNSS data file with the updated position and processing. This is typically only done once and removes unrealistic steps between position solutions.

Line 132: What is the 'reported processing uncertainty'?

The formal position uncertainty is estimated by the NRCAN CSRS-PPP service.

Line 133: More detail is required here to understand the processing steps and error propagation.

We think this is clear. The gradient in the x direction, dx/dt, provides the velocity in the x direction. The uncertainty in this gradient provides the uncertainty in  $v_x$ . The same goes for the gradient in the y direction. We propagate the uncertainties by adding percentage uncertainties when determining the total velocity magnitude and direction by adding the percentage uncertainties when multiplying and dividing.

Line 136: Is aliasing the correct word here?

We think aliasing is fine here.

Line 141: Where is this presented?

Figures 3 and 5, this has been edited in the updated manuscript.

Line 159: Is melt varied across the whole domain or locally? Assuming these units are m/a (velocity) per m3/a (melt), it is not clear over what area of the ice shelf the melt is integrated.

Yes the melt perturbation  $\delta \dot{M}_b$  is spatially variable (hence the use of a directional derivative, and Automatic Differentiation), and is expressed in m/a. The integral is over  $\Omega$  the entire model domain (although the gradient will be 0 over grounded ice since melt is not applied there).

Line 181: It would be useful to know what the baseline MITgcm basal melt rates are and how / if they vary seasonally before the perturbation is applied.

A figure showing the baseline MITgcm basal melt rates will be added to the Appendix of the updated manuscript. In this figure, we can see that the basal melt rates do vary seasonally, with a peak observed in the austral summer. However, we also see that there is a lot of noise in the MITgcm basal melt rates throughout the year, with multiple smaller peaks shown in the austral winter.

Fig 2 caption: The whole of the RIS and islands are also outlined in black so it is not clear which bit is 'passive ice'.

Passive ice will be identified using a different colour in the updated manuscript. The caption will be edited accordingly.

Fig 3. What is meant by 'errors'?

The errors are described in Section 2.2 as uncertainties. This will be edited in the Figure 3 caption.

Line 247: I don't see evidence that 'local changes in basal melt influence the velocities at Site 4'? This also seems to contradict the conclusions (Line 486).

In Figure 4 Site 4 we see high sensitivity at the KIS grounding zone to changes to basal melt. Here we are highlighting that local changes in basal melt AT the KIS grounding zone impact the velocities at Site 4. In the conclusions, we suggest that these local changes in perturbed basal melt rates AT the KIS grounding zone are unrealistic and thus changes in basal melt do not influence the velocities at Site 4. We can clarify this in the text in the updated manuscript.

Line 265: What is the difference? It could be important to note here.

The aim of this paper is not to try and match the GNSS and modelled velocities, but rather we are trying to determine whether it is possible to reproduce similar velocity variations from basal melt. We want to provide a proof of concept and test our hypothesis that perturbations (both in magnitude and phase) in basal melting at the identified sensitive regions could reproduce a similar behaviour in velocities to the GNSS observations. Therefore we will not quantify the difference, as we deem this is outside our aims of the study.

Figure 6: Is this the baseline dataset (i.e. perturbation of 0 m/a)? In which case why is there no velocity variability in the 0 m/a perturbation scenario (Fig 5). The mean and max imply seasonality in this baseline dataset.

Yes, this is the baseline dataset, and a time series of the baseline dataset will be included in the updated manuscript. We observed no velocity variability in the 0 m/a perturbation scenario, and we suggest this is due to the high seasonal basal melt values shown in Figure 5 not occurring at the GNSS sites locations, but close by (especially for Site 2). Hence, why the perturbations in basal melt rates were needed in our experiments.

Line 422: I don't think it is justified to say that "the majority of Site 2's intra-annual velocity variability is driven by seasonal changes in melting".

This has been changed to "Site 2's intra-annual velocity variability could partly be driven by seasonal changes in melting".

Line 449: Duplicate reference of Liu and Miller, 1979.

This has been removed.

Line 456: What is the mechanism for a short period of surface melting to lead to a velocity increase in the centre of the ice shelf, far from any shear zone?

This paragraph will be reworded, and sentence removed, as it does seem unlikely that a short period of surface melting in the centre of the ice shelf can lead to a velocity increase. However, Site 3 is located near one of the major rifts on the RIS, where shear stresses could be impacted by surface melting leading to velocity variations. The impact of the El Nino event on the basal melt rates will instead be focused on. [Klein et al., 2020] state that the surface heat fluxes over the ocean during this surface melt event in January 2016 may have been substantially different than those used to drive the ocean model that provided the basal melt rates used to force their ice-flow model. This will also be further discussed in our updated manuscript.

Line 460: This is a misrepresentation of what is stated in Mosbeux et al., (2023) who do tentatively attribute the 6-monthly signal to tides.

This sentence will be edited, and further discussion will be provided on the findings of [Mosbeux et al., 2023] in the updated version of the manuscript.

**References**

- Baldacchino, F., M. Morlighem, N. R. Golledge, H. Horgan, and A. Malyarenko, Sensitivity of the ross ice shelf to environmental and glaciological controls, *The Cryosphere*, 16(9), 3723–3738, doi:10.5194/tc-16-3723-2022, 2022.
- Fürst, J. J., G. Durand, F. Gillet-Chaulet, L. Tavard, M. Rankl, M. Braun, and O. Gagliardini, The safety band of Antarctic ice shelves, *Nature Climate Change*, 6(5), 479–482, doi:10.1038/nclimate2912, 2016.

- Gudmundsson, G. H., F. S. Paolo, S. Adusumilli, and H. A. Fricker, Instantaneous antarctic ice sheet mass loss driven by thinning ice shelves, *Geophysical Research Letters*, 46(23), 13,903–13,909, 2019.
- Klein, E., C. Mosbeux, P. D. Bromirski, L. Padman, Y. Bock, S. R. Springer, and H. A. Fricker, Annual cycle in flow of Ross Ice Shelf, Antarctica: Contribution of variable basal melting, *Journal of Glaciology*, 66(259), 861–875, doi:10.1017/jog.2020.61, 2020.
- Mosbeux, C., L. Padman, E. Klein, P. Bromirski, and H. Fricker, Seasonal variability in antarctic ice shelf velocities forced by sea surface height variations, *The Cryosphere*, 17(7), 2585–2606, 2023.
- Reese, R., G. H. Gudmundsson, A. Levermann, and R. Winkelmann, The far reach of ice-shelf thinning in Antarctica, *Nature Climate Change*, 8(1), 53–57, doi:10.1038/s41558-017-0020-x, 2018.
- Stewart, C. L., P. Christoffersen, K. W. Nicholls, M. J. Williams, and J. A. Dowdeswell, Basal melting of Ross Ice Shelf from solar heat absorption in an ice-front polynya, *Nature Geoscience*, 12(6), 435–440, doi:10.1038/s41561-019-0356-0, 2019.

---

## Author Comment (AC3)

**Modelling GNSS-observed seasonal velocity changes of the Ross Ice Shelf, Antarctica, using the Ice-sheet and Sea-level System Model (ISSM)**
**– Response to Reviewer 3 –**

Francesca BALDACCHINO et al

March 15, 2024

Firstly, we would like to thank all three reviewers for their constructive and detailed comments. We agree with many points that were raised, especially the lack of discussion of our results in the context of recently published papers (e.g., *Klein et al.* [2020] & *Mosbeux et al.* [2023]). We have responded to each reviewer's comments below. All three reviewers' main comments included the need for additional discussion and consideration of *Klein et al.* [2020] and *Mosbeux et al.* [2023]. This is a good point that we will address throughout a revised manuscript, as we realize that we did not adequately motivate and contextualise our study, and these previous works deserved more recognition.

**Novel contributions**

Several reviewers questioned the novelty of aspects of our study. Here we briefly summarise aspects of our study which we believe are novel contributions.

- We present new GNSS time series from the Ross Ice Shelf that have not previously been published. These include sites near the calving front, near a significant pinning point, and in the deep interior of the ice shelf near the grounding line.

- Notably, our Site 2 is close to the calving front near the Ross Island region, which has been identified as observing high basal melt rates on a seasonal timescale (Stewart et al., 2019).

- We show that these measurements consistently show 2 peaks in ice shelf velocity every year (for Sites 1, 2 and 4, the newly collected datasets), contrary to previous measurements presented in *Klein et al.* [2020] and *Mosbeux et al.* [2023].

- We suggest that the seasonal variability of SSH (i.e., yearly cycle) may not be able to reproduce our GNSS seasonal velocity variability (i.e., semi-annual).

- We therefore turned to the potential role of basal melt and wanted to test *what it would take* to match velocity variations by changing the forcing as little as possible.

- Our approach of combining Automatic Differentiation and weekly MITgcm basal melt rates ([*Klein et al.*, 2020] used monthly basal melt rates) is also novel.

**Sea surface height (SSH)**

An area where the two previous studies should be discussed more in our manuscript is in regards to what other factors that could be driving the observed velocity variations on the ice shelf. [*Mosbeux et al.*, 2023] nicely shows that the seasonal variability of SSH can explain their observed seasonal variability of ice velocity.

To take this into account, we will rerun our simulations with the same SSH forcings implemented in [*Mosbeux et al.*, 2023] to consider this factor. However, we expect that the seasonal variability of SSH cannot explain our two-peaked seasonal velocity variability, as mentioned above. In our revised manuscript, we will also discuss other possible factors (tides, sea ice buttressing etc), that may also be good candidates to explain our new GNSS observations.

**Basal melt rates**

Melt rates are difficult to model and properly constrain, especially close to grounding lines, despite their critical role on ice dynamics. All reviewers commented on the realism of the basal melt rates. We agree that these basal melt rate perturbations we use are extremely high for the Ross Ice Shelf, today and in the future. However, this paper focuses on asking whether perturbations in basal melt rates *can* reproduce a similar velocity variability as observed by the GNSS units. We acknowledge that our contribution is a proof of concept, not a definitive answer to the question, and we will do our best to make this clear in the revision.

**Multiple peaks in melt rate perturbation**

Several reviewers questioned our use of multiple peaks in melt rate perturbation. Here we clarify our motivation for doing so. The baseline weekly MITgcm basal melt rates include a clear peak in the austral summer, and multiple other (much smaller) peaks throughout the year, highlighting that the basal melt rates have more variability than presented in [*Klein et al.*, 2020]. We also refer to [*Stewart et al.*, 2019] basal melt observations in our discussion, highlighting that they observe the largest peak in the austral summer, but also smaller peaks in the austral winter.

*Klein et al.* [2020] suggest that the actual total summer increase in the heat content of the AASW layer near the ice front is likely to be larger than the modelled increase, and the seasonal enhancement of the basal melting will continue further into autumn than in their model. [*Klein et al.*, 2020] extended the late melt period to April and found that it also shifted the timing of maximum velocity a month later, showing that a longer or later melt period at the front could align the modelled and observed velocity phases.

Our approach is to use multiple basal melt peaks as the basis for our phasing of the basal melt forcing, and we apply perturbations on this forcing until we reproduce a similar velocity variability to the GNSS observations. Through this, we can highlight that seasonal basal melt rates can reproduce the GNSS velocity variability on an interannual timescale for XX of the sites. We do not state or intend to imply that these perturbed basal melt rates are realistic for the Ross Ice Shelf. Our study instead serves as a proof of concept, motivated by Klein et al. "as-yet-unidentified seasonal processes". This overall aim will be clarified in the revised manuscript.

**1 Reviewer 3**

**1.1 General comments**

I therefore have several concerns regarding the realism of the modeled melt rates and the conclusions of the paper. Furthermore, the paper overlooks the potential influence of other factors such as sea surface height variations and tidal effects, which have been shown to significantly impact ice flow dynamics in previous research. Even focusing solely on basal melt rates, seasonal melt close to the grounding line where ocean models usually struggle to correctly model high melt rates (e.g., the melt under Pine Island ice shelf in Dutrieux et al., 2013) and their effect on the grounding zone, could have been explored by the authors

Thank you for this comment. Firstly, we are aware now that we did not discuss the potential influence of other factors in enough detail. This will be added in the updated manuscript, as well as discussing the [*Klein et al.*, 2020] and [*Mosbeux et al.*, 2023] studies in more detail. As detailed in our summary response above, we will take into account sea surface height variations and discuss tidal effects, when concluding the influence of basal melting on the observed velocity variations. Regarding the realism of the modelled melt rates and the conclusion of the paper, please also refer to our summary response above.

**1.2 Specific comments**

Figure 1: To me, this figure could be reworked and made cleaner. Why drawing null velocities in the ocean? It only decreases the readability

Figure 1 will be reworked and made cleaner in the updated manuscript.

On site 3, which is the main site used by [*Klein et al.*, 2020], the data derivation from displacement to velocities gives you a minimum in April.

Yes, this is correct. These are the velocities we obtained using our processing steps as outlined in the manuscript, and our results compare well with [*Klein et al.*, 2020]. We show similar seasonal variability in the GNSS-derived velocities at Site 3, however, we do observe a minimum in April and a maximum in August, which are offset by 1 month compared to results presented in [*Klein et al.*, 2020] (minimum in March and maximum in July).

Figure 2. The figure really looks like a draft and not a publishable figure. The grounding line and the safety bands are both plotted in black. There is no metrics on the x and y that are used and written. The southern part of the grounding line is cutoff without specific reasons

We agree with the reviewer that this figure needs work, but following the recommendation from reviewer #1, we will instead remove this figure in the updated manuscript as suggested by Reviewer 1.

To me, if the MITgcm modelling shows a seasonality in melt rates, this seasonality should be explored, even if it does not give the correct phasing on the ice flow velocities. The MITgcm melt rates should be shown with maps of melt rates at different period of the years, or at least with a timeseries of the integrated melt rates over the ice shelf. For example, the model melt rates in [*Klein et al.*, 2020] shows only one peak melt rate in February (see their Figure 7a or the maps in Fig. 8). Why building a twice peaking melt rate if it is not realistic or backed by any modelling or observation?

A figure showing the timeseries of the baseline MITgcm basal melt rates will be added to the updated manuscript. As detailed in our above responses, we are using these perturbations in basal melt rates to understand whether basal melting *CAN* reproduce the GNSS observed velocity variations, assuming that MITgcm basal melt rates are imperfect and may not include all possible variability (especially in the vicinity of grounding lines).

Figure 5: Looking at the pattern of your observed velocity variations, it seems that ice flow reaches a minimum velocity in March and a second one in August. My understanding is that this is the reason why the authors apply two peak melt rates in your idealized sinusoidal melt. However, such a semi-annual cycle caused by something different like a semi-annual variability in tidal amplitudes and affecting the grounding zone of the ice shelf, as suggested in [*Mosbeux et al.*, 2023] conclusions. This could be seen as a process similar to the nonlinear response of the ice shelf (and the ice sheet) to the diurnal tide (e.g. Gudmnundsson, 2011; Rosier et al., 2020). Site 3 semi-annual cycle does not seem as clean as on other sites but still visible with a sharp drop in velocity in November followed by plateau from early January to March, a second drop in March-April before a reversal with a speed up until August, ending with a second Plateau from August to November. From the detrended displacement in Figure A4, we do not see any sharp change in displacement in November. How do you explain such result? Also, the strong direction changes before January 2016, does not really reflect in the detrended x and y displacement. Looking at [*Klein et al.*, 2020], the velocity trend looks a bit different. It would be good to investigate the reasons for this.

This is an interesting point, we will include further discussion about other potential factors (SSH and tides) that could be driving the observed variability in velocities in the updated manuscript. The differences between our velocity trend and [*Klein et al.*, 2020] velocities will be discussed in more detail in the updated manuscript. The difference could be due to [*Klein et al.*, 2020] using T-TIDE analyses to remove the tidal signals in the dataset, and could also be due to the time window used. In this paper, we use a time window of 8 weeks to smooth the short-term tidal effects and to identify seasonal changes.

**References**

Klein, E., C. Mosbeux, P. D. Bromirski, L. Padman, Y. Bock, S. R. Springer, and H. A. Fricker, Annual cycle in flow of Ross Ice Shelf, Antarctica: Contribution of variable basal melting, *Journal of Glaciology*, *66*(259), 861–875, doi:10.1017/jog.2020.61, 2020.

Mosbeux, C., L. Padman, E. Klein, P. Bromirski, and H. Fricker, Seasonal variability in antarctic ice shelf velocities forced by sea surface height variations, *The Cryosphere*, *17*(7), 2585–2606, 2023.

Stewart, C. L., P. Christoffersen, K. W. Nicholls, M. J. Williams, and J. A. Dowdeswell, Basal melting of Ross Ice Shelf from solar heat absorption in an ice-front polynya, *Nature Geoscience*, *12*(6), 435–440, doi:10.1038/s41561-019-0356-0, 2019.

---

## Referee Report (RR1)

**Modelling GNSS-observed seasonal velocity changes of the Ross Ice Shelf, Antarctica, using the Ice-sheet and Sea-level System Model (ISSM)**

I first want to apologize for the delay in this second review. I partially delayed it because I wanted to make sure it was fair and right. As I explain it hereafter in greater details, I found this work interesting and definitely useful to the community but I think that additional warnings are needed when it comes to the conclusions.

I thank the authors for their answer to most of my questions about the manuscript and their considerations for my suggestions and suggestions by other reviewers, especially the new simulations including SSH perturbations. I think that the manuscript has largely improved although, in my opinion, the conclusions on the melt-induced seasonal velocity changes remain too strong and misleading. The authors should only claim what it can: they apply a synthetical basal melt to get the desired ice flow response and show that they succeed… but that the necessary synthetical basal melt is very far from realistic, given state of the art ocean modelling and subsequent basal melt estimations. Here after, I expose my main arguments (given my understanding of the paper).

The sinusoidal perturbation applied to the basal melt have been argued be too strong during the previous review. Such perturbation was indeed surprising given the low melt rates typically observed over the region, as rightly reminded by the authors: *"The RIS basal melt rates are relatively low due to the cold dense water masses formed on the continental shelf [...]"*. The authors justify this choice based on observations by Steward et al. (2019): "*Recently, high basal melt rates have been observed at the calving front near Ross Island due to the seasonal inflow of summer-warmed AASW from the adjacent Ross Sea Polynya downwelling into the ice shelf cavity*". However, this high seasonal melt has been observed close the calving front and Ross Island. Previous modelling has shown that this intense melt rate is seasonal (e.g., Tinto et al., 2019) and occurs over January-March, which coincides with observations (e.g., Steward et al., 2019).

My issue, as I stated in my first review, is that the authors choose to (1) synthetically introduce melt at specific spots guided by the sensitivity map to maximize the impact of the forcing with (2) a timing guided by a sinusoid specifically phased to match the velocity change observed, and (3) set the amplitude of the sinusoid to values largely exceeding MITgcm values. I quickly tried to compare the MITgcm melt to the sinusoid used for perturbation–notice that I could be wrong since I did this by hand (to me, this figure combination would be a very nice addition to the paper)–and I get the following figure, where we see a very different phasing of the two components:

[Figure]

I agree that ocean models have their limitations but correctly modelling melt rates have been one of the main goals in the community interested in ice-ocean interaction over the last decades, with great improvement and melt rates matching observations to a great extent. MITgcm is one of the leading ocean models in this matter and is, to me, unlikely to be off by such an extent. When it comes to Ross Ice Shelf, the comparison of the modelling by Tinto et al. (2019) and the matching with observations by Steward et al. (2019) is a striking example of the current capacity of state-of-the-art models. In any case, the speed changes induced by the MITgcm melt rates are neglectable (blue line in Figure 4) with respect to simulations including the sinusoidal perturbation, meaning that either MITgcm is significantly wrong or that melt rates are not a key factor until we apply perturbations that do not look at all like MITgcm. In this regard, the phasing of SSH-induced speed change seems to be off too (I understand that you apply the same forcing as Mosbeux et al. (2023) but only with a hydrostatic grounding line migration) but the amplitude of the changes looks better than the melt rate induced perturbation without requiring a synthetical tuning. The comparison is also not entirely fair since you compare melt rates applied to specifically sensitive regions to realistic SSH over the entire domain.

In conclusion, at the reading of the paper, it seems that the authors pushed a bit too far their storyline, i.e., "melt-induced seasonal speed change". The conclusions of the paper should be tempered before publication and the synthetical approach applied to build the ocean forcing should be better emphasized in the discussion. I strongly believe that the future of ice sheet modelling relies in matching such seasonal variations and that this manuscript shows valuable results in this context, i.e., melt rates is a factor but maybe not the main factor to model such speed changes if we trust the ocean model (which I personally do more than a forcing tuned to match a target). This conclusion aligns well with Klein et al. (2020) and I think it is a good thing to have two studies using slightly different approaches to get to a similar conclusion. The number of adjustments on the model melt rates required to reproduce the data is just too high. *"When a measure becomes a target, it ceases to be a good measure" (Goodhart's Law).*

I also have a few specific questions concerning the revised version.

**Specific comments**

In the abstract, we can read: "*While this study does not bring a definitive answer to the question of what the drivers of seasonality in ice flow are, this study shows that seasonal basal melt rates could explain the GNSS velocity variability on an* (typo) *seasonal timescale for two of the four GNSS sites*". This is an example of what I would call "too strong" or "misleading" conclusions due to the forcing needed.

I thank the authors for reworking their figures based on some comments, especially Figure 1. However, to me, Figure 3 does not look good enough. I do not understand why the grounding line is randomly cut-off on the left and on the top of the panels (same for Figure A5, A6 and A7). Figure 5, suffers the same issue and personally think that melt on the grounded ice and especially on the open ocean should not be displayed as a null value but just left "blank".

Figure A3: the y axis for the direction has a strange unit system (e.g., 0.1 +1.896e2 degrees is not very reader friendly)

Figure A9: Why is the time going from 2040 to 2042? It would be interesting to align A8 and A9 to compare MITgcm melt rates in sensitive regions (see my earlier suggestion). Also, what is the meaning of the different colors and lines?

Line 669: "exploring" instead of "exploration of" to stay consistent with (1) and (2)?

Line 673: I think that this correction, explaining that the perturbations are large (maybe add that the design of the experiment leads to make the perturbation in very sensitive spot, which is not necessarily the case with real melting). I think that you should ensure that this level of scientific restraint is maintained throughout the entire manuscript.

---

## Referee Report (RR2)

**Modelling GNSS-observed seasonal velocity changes of the Ross Ice Shelf, Antarctica, using the Ice-sheet and Sea-level System Model (ISSM)**

I thank the authors for revisiting their study and tempering their conclusions. I believe the refocused discussion and conclusions now align more closely with their results and with other research on this topic. I have a few minor suggestions that I would like to submit to the authors for consideration.

**Specific comments and (mainly) suggestions**

*Line numbers correspond to the "Author's tracked changes" version*

- Line 3: the comma placement seems a bit unusual; please consider revising.
- Lin 11: Suggested wording "Here, we investigate the potential role of basal melt variability on ice flow speed and use the Ross Ice Shelf a testbed"
- Line 20-21: consider rephrasing to "… on ice flow speed, the amplitude of the perturbation required to ..."
- Line 61: replace ";" with "and"
- Line 209: be sure to comment on the unrealistic refreezing that is applied in the sine function.
- Line 410: consider specifying what you mean by "… as has been done previously". For example, in Reese et al. (2018) they also look at a local perturbation.
- Line 420-425: I suggest to better link the two with something like: "This can be explained by the loss of buttressing force triggered by the ice shelf thinning."

  The explanation of the impact of thinning on pinning points could be more detailed. Depending on the way you apply the melting, you may not lose any grounded area over Ross Island in your experiments. The loss of buttressing would therefore be due to a reduced thickness close to the pinning point and a decreased transmission of stresses from the pinning point to the rest of the ice shelf. This might not be clear in the text right now, as the reader might think that melting leads to a reduction of grounded area (which might be the case though).

  If there is some unpinning, I think that it will only occur in ISSM if you apply melting at the first grounded node or if the thinning over the ice floating area leads to an increased advection of ice from a grounded area to a floating area (which, again, might be the case). This might be good to check in the outputs and to emphasize in the discussion.

- Line 510: This sentence nicely recaps the limitation of the experiments but is a bit long. Here is a suggestion to improve readability: "However, we highlight that if melt alone were responsible, and it occurred only in sensitive regions of the ice shelf, then a variability in basal melting with peaks of 20-80 m/a in April and October would be needed to match the GNSS observations at sites 1, 2 and 4."
- Line 580: As in line 420-425, it might be good to specify the effect of ice shelf melting on pinning points.
- Line 588: "Therefore" is repeated from line 585. Consider rephrasing to avoid redundancy.
- Line 591: Remove the period at the beginning.

- **Specific comment on previous review:**

In the previous review I asked the following question: Figure A9: Why is the time going from 2040 to 2042? It would be interesting to align A8 and A9 to compare MITgcm melt rates in sensitive regions. Also, what is the meaning of the different colours and lines?

Which has been answered as follows: We chose 2040-2042 arbitrarily. This figure provides an example of the sinusoidal pattern, which is repeated throughout the years. We have decided not to include this figure, as we think A8 and A9 highlight the differences in phase well without needing to spend significant time combining the figures. Each coloured line represents basal melt at a different sensitive region (i.e., each node on the model mesh).

Thank you for the response. However, I believe combining Figures A8 and A9 or aligning their time axes would be highly be highly beneficial to readers, as it would allow for straightforward comparison of MITgcm with the sinusoidal forcing. This seems logical to me and is little extra-work but I leave this decision to the authors.

Also, in your response, you mention that each color on the plot represents basal melt at different nodes of the mesh. Could you please include this information in the figure label for clarity?

---

## Author Response (AR2)

Firstly, we would like to thank all reviewers for their constructive and detailed comments.

**Main comments**

We understand the main concerns that were raised regarding the conclusions being too 'strong'. We have addressed this issue throughout our revised manuscript as follows:

- We have removed or toned down some of the conclusions and claims. In particular we have focused on editing these statements in the abstract, discussion and conclusion.

- We have restructured our aims to clarify that we use the RIS as a *testbed* to see if local changes in basal melt can affect flow speed, and what magnitude of variability is needed to match the GNSS observed velocity changes.

- We have restructured the Discussion to include more details on our methodology limitations and caveats, the type of perturbations used and comparison with observations. Additionally, we have removed our strong statements regarding MITgcm ocean model and the SSH results.

- We have highlighted throughout that our basal melt perturbations are likely not realistic for the RIS because of (1) the high magnitudes and (2) the sinusoidal pattern, and we suggest that other mechanisms are also at play to drive the observed velocity changes.

- We focus on discussing the sensitivity maps and how these results show that localized changes in melt can have a strong impact on flow speed in general.

We highlight again that our study brings novel contributions, which include:

- We present new GNSS time series from the Ross Ice Shelf that have not previously been published. These include sites near the calving front, near a significant pinning point, and in the deep interior of the ice shelf near the grounding line.

- We show that these measurements consistently show 2 peaks in ice shelf velocity every year (for Sites 1, 2 and 4, the newly collected datasets), contrary to previous measurements presented in *Klein et al.* [2020] and *Mosbeux et al.* [2023].

- We use the RIS as a *testbed* to see if local changes in basal melt can affect flow speed, and what magnitude of variability is needed to match the GNSS observed velocity changes.

- We use a novel approach of combining Automatic Differentiation and weekly MITgcm basal melt rates ([*Klein et al.*, 2020] used monthly basal melt rates).

- Our final sensitivity maps allow us to understand that localized changes in melt can have a strong impact on ice flow on the RIS.

We have responded to the reviewer's specific comments below.

**Reviewer 1: Specific comments**

However, to me, Figure 3 does not look good enough. I do not understand why the grounding line is randomly cut off on the left and the top of the panels (same for Figures A5, A6 and A7). Figure 5, suffers the same issue and I personally think the melt on the grounded ice and open ocean should not be displayed as null value, but just left blank

Thank you for pointing this out. Figures 3, A5, A6 and A7 have been edited to make sure the grounding line is not cut off. We decided to remove Figure 5.

Figure A3: the y axis for the direction has a strange unit system.

Figure A3 has been edited so that the y-axis has a user friendly unit system.

*Figure A9: Why is the time going from 2040 to 2042? It would be interesting to align A8 and A9 to compare MITgcm melt rates in sensitive regions. Also, what is the meaning of the different colours and lines?*

We chose 2040-2042 arbitrarily. This figure provides an example of the sinusoidal pattern, which is repeated throughout the years. We have decided not to include this figure, as we think A8 and A9 highlight the differences in phase well without needing to spend significant time combining the figures. Each coloured line represents basal melt at a different sensitive region (i.e., each node on the model mesh).

*Line 669: 'exploring' instead of 'exploration of' to stay consistent with (1) and (2)?*

Done.

**Reviewer 2: Specific comments**

*The comment from the previous review has not been addressed: 'It is not inconceivable that small, solar annual or semi-annual tides could drive the remaining 1 percent semi-annual variations in velocity shown in Figure 3 and it needs to be explained why they can be ignored.*

This has been addressed in Discussion section titled: Potential drivers of intra-annual velocity variation.

*Now that you have added a figure on MITgcm model baseline melt rates, I find it strange that while there is one very dominant seasonal peak in Jan, there appears to be no velocity variation in your unperturbed run. If seasonal melt rates lead to velocity change, surely this should exist in the unperturbed state due to the January peak?*

The MITgcm baseline melt rates drive very very small velocity variations ranging from -0.05 to 0.05 m/a at the GNSS sites. Therefore, we perturbed (in both phase and amplitude) the MITgcm basal melt rates significantly to match the GNSS observations.

*It also is not mentioned in the text that your perturbation involves a very high negative melt rate (Figure A9). This is perhaps even more unrealistic than the positive melt rates.*

Thank you for highlighting this. This has been clarified in the Methodology and Discussion sections.

**References**

Klein, E., C. Mosbeux, P. D. Bromirski, L. Padman, Y. Bock, S. R. Springer, and H. A. Fricker, Annual cycle in flow of Ross Ice Shelf, Antarctica: Contribution of variable basal melting, *Journal of Glaciology*, *66*(259), 861–875, doi:10.1017/jog.2020.61, 2020.

Mosbeux, C., L. Padman, E. Klein, P. Bromirski, and H. Fricker, Seasonal variability in antarctic ice shelf velocities forced by sea surface height variations, *The Cryosphere*, *17*(7), 2585–2606, 2023.

---

## Author Response (AR3)

**Modelling GNSS-observed seasonal velocity changes of the Ross Ice Shelf, Antarctica, using the Ice-sheet and Sea-level System Model (ISSM)**
**– Response to Reviewer –**

Francesca BALDACCHINO et al

November 13, 2024

Firstly, we would like to thank the reviewer for their final supportive and constructive suggestions. We have responded to the reviewer's specific comments below.

**1 Reviewer Suggestions**

Line 3: the comma placement seems a bit unusual; please consider revising.

Done. We have changed the sentence to: 'However, the drivers of this variability remain poorly understood'.

Line 11: Suggested wording "Here, we investigate the potential role of basal melt variability on ice flow speed and use the Ross Ice Shelf a testbed"

Done.

Line 20-21: consider rephrasing to "... on ice flow speed, the amplitude of the perturbation required to ..."

Done.

Line 61: replace ";" with "and"

Done.

Line 209: be sure to comment on the unrealistic refreezing that is applied in the sine function.

Done. We comment on this in the Discussion section titled: 'Magnitude of Variability'.

Line 410: consider specifying what you mean by "... as has been done previously". For example, in Reese et al. (2018) they also look at a local perturbation.

Done. We have added in the reference *Klein et al.* [2020].

Line 420-425: I suggest to better link the two with something like: "This can be explained by the loss of buttressing force triggered by the ice shelf thinning."

Done. Added this sentence 'This can be explained by the loss of buttressing force triggered by ice shelf thinning near the Ross Island pinning point.'

The explanation of the impact of thinning on pinning points could be more detailed. Depending on the way you apply the melting, you may not lose any grounded area over Ross Island in your experiments. The loss of buttressing would therefore be due to a reduced thickness close to the pinning point and a decreased transmission of stresses from the pinning point to the rest of the ice shelf. This might not be clear in the text right now, as

the reader might think that melting leads to a reduction of grounded area (which might be the case though). If there is some unpinning, I think that it will only occur in ISSM if you apply melting at the first grounded node or if the thinning over the ice floating area leads to an increased advection of ice from a grounded area to a floating area (which, again, might be the case). This might be good to check in the outputs and to emphasize in the discussion.

Done. We have added this sentence into the discussion section titled 'Local Perturbations': 'This loss of buttressing force is due to reduced ice thickness near the Ross Island pinning point, and a decreased transmission of stresses from the pinning point to the rest of the ice shelf.'

Line 510: This sentence nicely recaps the limitation of the experiments but is a bit long. Here is a suggestion to improve readability: "However, we highlight that if melt alone were responsible, and it occurred only in sensitive regions of the ice shelf, then a variability in basal melting with peaks of 20-80 m/a in April and October would be needed to match the GNSS observations at sites 1, 2 and 4."

Done. We have included the suggested rephrased sentence.

Line 580: As in line 420-425, it might be good to specify the effect of ice shelf melting on pinning points.

Done. We have clarified this in the discussion section titled: 'Local Perturbations'.

Line 588: "Therefore" is repeated from line 585. Consider rephrasing to avoid redundancy.

Done. 'Therefore' was removed, and no longer repeated.

Line 591: Remove the period at the beginning.

Done.

However, I believe combining Figures A8 and A9 or aligning their time axes would be highly be highly beneficial to readers, as it would allow for straightforward comparison of MITgcm with the sinusoidal forcing. This seems logical to me and is little extra-work but I leave this decision to the authors.

We have decided not to include this figure, as it would be significant work to do so.

Also, in your response, you mention that each color on the plot represents basal melt at different nodes of the mesh. Could you please include this information in the figure label for clarity?.

Done.

**References**

Klein, E., C. Mosbeux, P. D. Bromirski, L. Padman, Y. Bock, S. R. Springer, and H. A. Fricker, Annual cycle in flow of Ross Ice Shelf, Antarctica: Contribution of variable basal melting, *Journal of Glaciology*, *66*(259), 861–875, doi:10.1017/jog.2020.61, 2020.